# Unexpected Improvements to Expected Improvement for Bayesian Optimization

**Sebastian Ament**
Meta
ament@meta.com

**Samuel Daulton**
Meta
sdaulton@meta.com

**David Eriksson**
Meta
deriksson@meta.com

**Maximilian Balandat**
Meta
balandat@meta.com

**Eytan Bakshy**
Meta
ebakshy@meta.com

## Abstract

Expected Improvement (EI) is arguably the most popular acquisition function in Bayesian optimization and has found countless successful applications, but its performance is often exceeded by that of more recent methods. Notably, EI and its variants, including for the parallel and multi-objective settings, are challenging to optimize because their acquisition values vanish numerically in many regions. This difficulty generally increases as the number of observations, dimensionality of the search space, or the number of constraints grow, resulting in performance that is inconsistent across the literature and most often sub-optimal. Herein, we propose LogEI, a new family of acquisition functions whose members either have identical or approximately equal optima as their canonical counterparts, but are substantially easier to optimize numerically. We demonstrate that numerical pathologies manifest themselves in "classic" analytic EI, Expected Hypervolume Improvement (EHVI), as well as their constrained, noisy, and parallel variants, and propose corresponding reformulations that remedy these pathologies. Our empirical results show that members of the LogEI family of acquisition functions substantially improve on the optimization performance of their canonical counterparts and surprisingly, are on par with or exceed the performance of recent state-of-the-art acquisition functions, highlighting the understated role of numerical optimization in the literature.

## 1   Introduction

Bayesian Optimization (BO) is a widely used and effective approach for sample-efficient optimization of expensive-to-evaluate black-box functions [25, 28], with applications ranging widely between aerospace engineering [48], biology and medicine [49], materials science [3], civil engineering [4], and machine learning hyperparameter optimization [66, 72]. BO leverages a probabilistic *surrogate model* in conjunction with an *acquisition function* to determine where to query the underlying objective function. Improvement-based acquisition functions, such as Expected Improvement (EI) and Probability of Improvement (PI), are among the earliest and most widely used acquisition functions for efficient global optimization of non-convex functions [42, 58]. EI has been extended to the constrained [27, 29], noisy [52], and multi-objective [20] setting, as well as their respective batch variants [6, 13, 77], and is a standard baseline in the BO literature [25, 66]. While much of the literature has focused on developing new sophisticated acquisition functions, subtle yet critical implementation details of foundational BO methods are often overlooked. Importantly, the performance of EI and its variants is inconsistent even for *mathematically identical* formulations and, as we show in this work, most often sub-optimal.

37th Conference on Neural Information Processing Systems (NeurIPS 2023).

Although the problem of optimizing EI effectively has been discussed in various works, e.g. [25, 31, 77], prior focus has been on optimization algorithms and initialization strategies, rather than the fundamental issue of computing EI.

In this work, we identify pathologies in the computation of improvement-based acquisition functions that give rise to numerically vanishing values and gradients, which – to our knowledge – are present in *all existing implementations of EI*, and propose reformulations that lead to increases in the associated optimization performance which often match or exceed that of recent methods.

**Contributions**

1. We introduce LogEI, a new family of acquisition functions whose members either have identical or approximately equal optima as their canonical counterparts, but are substantially easier to optimize numerically. Notably, the analytic variant of LogEI, which *mathematically* results in the same BO policy as EI, empirically shows significantly improved optimization performance.

2. We extend the ideas behind analytical LogEI to other members of the EI family, including constrained EI (CEI), Expected Hypervolume Improvement (EHVI), as well as their respective batch variants for parallel BO, qEI and qEHVI, using smooth approximations of the acquisition utilities to obtain non-vanishing gradients. All of our methods are available as part of BoTorch [6].

3. We demonstrate that our newly proposed acquisition functions substantially outperform their respective analogues on a broad range of benchmarks without incurring meaningful additional computational cost, and often match or exceed the performance of recent methods.

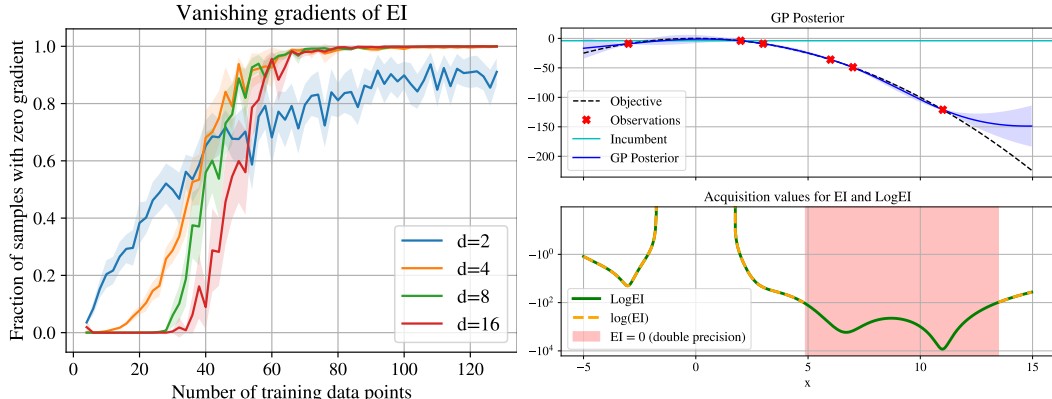

Figure 1: **Left:** Fraction of points sampled from the domain for which the magnitude of the gradient of EI vanishes to $< 10^{-10}$ as a function of the number of randomly generated data points $n$ for different dimensions $d$ on the Ackley function. As $n$ increases, EI and its gradients become numerically zero across most of the domain, see App. D.2 for details. **Right:** Values of EI and LogEI on a quadratic objective. EI takes on extremely small values on points for which the likelihood of improving over the incumbent is small and is numerically *exactly* zero in double precision for a large part of the domain ($\approx [5, 13.5]$). The left plot shows that this tends to worsen as the dimensionality of the problem and the number of data points grow, rendering gradient-based optimization of EI futile.

**Motivation**

Maximizing acquisition functions for BO is a challenging problem, which is generally non-convex and often contains numerous local maxima, see the lower right panel of Figure 1. While zeroth-order methods are sometimes used, gradient-based methods tend to be far more effective at optimizing acquisition functions on continuous domains, especially in higher dimensions.

In addition to the challenges stemming from non-convexity that are shared across acquisition functions, the values and gradients of improvement-based acquisition functions are frequently minuscule in large swaths of the domain. Although EI is never *mathematically* zero under a Gaussian posterior distribution,[1] it often vanishes, even becoming *exactly* zero in floating point precision. The same

---

[1]except at any point that is perfectly correlated with a previous noiseless observation

applies to its gradient, making EI (and PI, see Appendix A) exceptionally difficult to optimize via gradient-based methods. The right panels of Figure 1 illustrate this behavior on a simple one-dimensional quadratic function.

To increase the chance of finding the global optimum of non-convex functions, gradient-based optimization is typically performed from multiple starting points, which can help avoid getting stuck in local optima [70]. For improvement-based acquisition functions however, optimization becomes increasingly challenging as more data is collected and the likelihood of improving over the incumbent diminishes, see our theoretical results in Section 3 and the empirical illustration in Figure 1 and Appendix D.2. As a result, gradient-based optimization with multiple random starting points will eventually degenerate into random search when the gradients at the starting points are numerically zero. This problem is particularly acute in high dimensions and for objectives with a large range.

Various initialization heuristics have been proposed to address this behavior by modifying the random-restart strategy. Rather than starting from random candidates, an alternative naïve approach would be to use initial conditions close to the best previously observed inputs. However, doing that alone inherently limits the acquisition optimization to a type of local search, which cannot have global guarantees. To attain such guarantees, it is necessary to use an asymptotically space-filling heuristic; even if not random, this will entail evaluating the acquisition function in regions where no prior observation lies. Ideally, these regions should permit gradient-based optimization of the objective for efficient acquisition function optimization, which necessitates the gradients to be non-zero. In this work, we show that this can be achieved for a large number of improvement-based acquisition functions, and demonstrate empirically how this leads to substantially improved BO performance.

## 2 Background

We consider the problem of maximizing an expensive-to-evaluate black-box function $\mathbf{f}_{\text{true}} : \mathbb{X} \mapsto \mathbb{R}^M$ over some feasible set $\mathbb{X} \subseteq \mathbb{R}^d$. Suppose we have collected data $\mathcal{D}_n = \{(\mathbf{x}_i, \mathbf{y}_i)\}_{i=1}^n$, where $\mathbf{x}_i \in \mathbb{X}$ and $\mathbf{y}_i = \mathbf{f}_{\text{true}}(\mathbf{x}_i) + \mathbf{v}_i(\mathbf{x}_i)$ and $\mathbf{v}_i$ is a noise corrupting the true function value $\mathbf{f}_{\text{true}}(\mathbf{x}_i)$. The response $\mathbf{f}_{\text{true}}$ may be multi-output as is the case for multiple objectives or black-box constraints, in which case $\mathbf{y}_i, \mathbf{v}_i \in \mathbb{R}^M$. We use Bayesian optimization (BO), which relies on a surrogate model $\mathbf{f}$ that for any *batch* $\mathbf{X} := \{\mathbf{x}_1, \ldots, \mathbf{x}_q\}$ of candidate points provides a probability distribution over the outputs $f(\mathbf{X}) := (f(\mathbf{x}_1), \ldots, f(\mathbf{x}_q))$. The acquisition function $\alpha$ then utilizes this posterior prediction to assign an acquisition value to $\mathbf{x}$ that quantifies the value of evaluating the points in $\mathbf{x}$, trading off exploration and exploitation.

### 2.1 Gaussian Processes

Gaussian Processes (GP) [65] are the most widely used surrogates in BO, due to their high data efficiency and good uncertainty quantification. For our purposes, it suffices to consider a GP as a mapping that provides a multivariate Normal distribution over the outputs $f(\mathbf{x})$ for any $\mathbf{x}$:

$$f(\mathbf{x}) \sim \mathcal{N}(\boldsymbol{\mu}(\mathbf{x}), \boldsymbol{\Sigma}(\mathbf{x})), \qquad \boldsymbol{\mu} : \mathbb{X}^q \to \mathbb{R}^{qM}, \quad \boldsymbol{\Sigma} : \mathbb{X}^q \to \mathcal{S}_+^{qM}. \tag{1}$$

In the single-outcome ($M = 1$) setting, $f(\mathbf{x}) \sim \mathcal{N}(\mu(\mathbf{x}), \Sigma(\mathbf{x}))$ with $\mu : \mathbb{X}^q \to \mathbb{R}^q$ and $\Sigma : \mathbb{X}^q \to \mathcal{S}_+^q$. In the sequential ($q = 1$) case, this further reduces to a univariate Normal distribution: $f(\mathbf{x}) \sim \mathcal{N}(\mu(\mathbf{x}), \sigma^2(\mathbf{x}))$ with $\mu : \mathbb{X} \to \mathbb{R}$ and $\sigma : \mathbb{X} \to \mathbb{R}_+$.

### 2.2 Improvement-based Acquisition Functions

**Expected Improvement** For the fully-sequential ($q = 1$), single-outcome ($M = 1$) setting, "classic" EI [59] is defined as

$$\text{EI}_{y^*}(\mathbf{x}) = \mathbb{E}_{f(\mathbf{x})}\big[[f(\mathbf{x}) - y^*]_+\big] = \sigma(\mathbf{x}) \, h\left(\frac{\mu(\mathbf{x}) - y^*}{\sigma(\mathbf{x})}\right), \tag{2}$$

where $[\cdot]_+$ denotes the $\max(0, \cdot)$ operation, $y^* = \max_i y_i$ is the best function value observed so far, also referred to as the *incumbent*, $h(z) = \phi(z) + z\Phi(z)$, and $\phi, \Phi$ are the standard Normal density and distribution functions, respectively. This formulation is arguably the most widely used acquisition function in BO, and the default in many popular software packages.

**Constrained Expected Improvement**    *Constrained BO* involves one or more black-box constraints and is typically formulated as finding $\max_{\mathbf{x} \in \mathbb{X}} f_{\text{true},1}(\mathbf{x})$ such that $f_{\text{true},i}(\mathbf{x}) \leq 0$ for $i \in \{2, \ldots, M\}$. Feasibility-weighting the improvement [27, 29] is a natural approach for this class of problems:

$$\text{CEI}_{y^*}(\mathbf{x}) = \mathbb{E}_{\mathbf{f}(\mathbf{x})} \left[ [f_1(\mathbf{x}) - y^*]_+ \prod_{i=2}^{M} \mathbb{1}_{f_i(\mathbf{x}) \leq 0} \right], \tag{3}$$

where $\mathbb{1}$ is the indicator function. If the constraints $\{f_i\}_{i \geq 2}$ are modeled as conditionally independent of the objective $f_1$ this can be simplified as the product of EI and the probability of feasibility.

**Parallel Expected Improvement**    In many settings, one may evaluate $f_{\text{true}}$ on $q > 1$ candidates in parallel to increase throughput. The associated parallel or batch analogue of EI [30, 75] is given by

$$\text{qEI}_{y^*}(\mathbf{X}) = \mathbb{E}_{f(\mathbf{X})} \left[ \max_{j=1,\ldots,q} \{ [f(\mathbf{x}_j) - y^*]_+ \} \right]. \tag{4}$$

Unlike EI, qEI does not admit a closed-form expression and is thus typically computed via Monte Carlo sampling, which also extends to non-Gaussian posterior distributions [6, 75]:

$$\text{qEI}_{y^*}(\mathbf{X}) \approx \sum_{i=1}^{N} \max_{j=1,\ldots,q} \{ [\xi^i(\mathbf{x}_j) - y^*]_+ \}, \tag{5}$$

where $\xi^i(\mathbf{x}) \sim f(\mathbf{x})$ are random samples drawn from the joint model posterior at $\mathbf{x}$.

**Expected Hypervolume Improvement**    In multi-objective optimization (MOO), there generally is no single best solution; instead the goal is to explore the Pareto Frontier between multiple competing objectives, the set of mutually-optimal objective vectors. A common measure of the quality of a finitely approximated Pareto Frontier $\mathcal{P}$ between $M$ objectives with respect to a specified reference point $\mathbf{r} \in \mathbb{R}^M$ is its *hypervolume* $\text{HV}(\mathcal{P}, \mathbf{r}) := \lambda(\bigcup_{\mathbf{y}_i \in \mathcal{P}} [\mathbf{r}, \mathbf{y}_i])$, where $[\mathbf{r}, \mathbf{y}_i]$ denotes the hyper-rectangle bounded by vertices $\mathbf{r}$ and $\mathbf{y}_i$, and $\lambda$ is the Lebesgue measure. An apt acquisition function for multi-objective optimization problems is therefore the expected hypervolume improvement

$$\text{EHVI}(\mathbf{x}) = \mathbb{E}_{\mathbf{f}(\mathbf{X})} \big[ [\text{HV}(\mathcal{P} \cup \mathbf{f}(\mathbf{X}), \mathbf{r}) - \text{HV}(\mathcal{P}, \mathbf{r})]_+ \big], \tag{6}$$

due to observing a batch $\mathbf{f}(\mathbf{X}) := [\mathbf{f}(\mathbf{x}_1), \cdots, \mathbf{f}(\mathbf{x}_q)]$ of $q$ new observations. EHVI can be expressed in closed form if $q = 1$ and the objectives are modeled with independent GPs [80], but Monte Carlo approximations are required for the general case (qEHVI) [13].

## 2.3   Optimizing Acquisition Functions

Optimizing an acquisition function (AF) is a challenging task that amounts to solving a non-convex optimization problem, to which multiple approaches and heuristics have been applied. These include gradient-free methods such as divided rectangles [41], evolutionary methods such as CMA-ES [32], first-order methods such as stochastic gradient ascent, see e.g., Daulton et al. [15], Wang et al. [75], and (quasi-)second order methods [25] such as L-BFGS-B [10]. Multi-start optimization is commonly employed with gradient-based methods to mitigate the risk of getting stuck in local minima. Initial points for optimization are selected via various heuristics with different levels of complexity, ranging from simple uniform random selection to BoTorch's initialization heuristic, which selects initial points by performing Boltzmann sampling on a set of random points according to their acquisition function value [6]. See Appendix B for a more complete account of initialization strategies and optimization procedures used by popular implementations. We focus on gradient-based optimization as often leveraging gradients results in faster and more performant optimization [13].

Optimizing AFs for parallel BO that quantify the value of a batch of $q > 1$ points is more challenging than optimizing their sequential counterparts due to the higher dimensionality of the optimization problem – $qd$ instead of $d$ – and the more challenging optimization surface. A common approach to simplify the problem is to use a *sequential greedy* strategy that greedily solves a sequence of single point selection problems. For $i = 1, \ldots, q$, candidate $\mathbf{x}_i$ is selected by optimizing the AF for $q = 1$, conditional on the previously selected designs $\{\mathbf{x}_1, \ldots, \mathbf{x}_{i-1}\}$ and their unknown observations, e.g. by fantasizing the values at those designs [77]. For submodular AFs, including EI, PI, and EHVI, a sequential greedy strategy will attain a regret within a factor of $1/e$ compared to the joint optimum, and previous works have found that sequential greedy optimization yields *improved* BO performance compared to joint optimization [13, 77]. Herein, we find that our reformulations enable joint batch optimization to be competitive with the sequential greedy strategy, especially for larger batches.

## 2.4 Related Work

While there is a substantial body of work introducing a large variety of different AFs, much less focus has been on the question of how to effectively implement and optimize these AFs. Zhan and Xing [81] provide a comprehensive review of a large number of different variants of the EI family, but do not discuss any numerical or optimization challenges. Zhao et al. [82] propose combining a variety of different initialization strategies to select initial conditions for optimization of acquisition functions and show empirically that this improves optimization performance. However, they do not address any potential issues or degeneracies with the acquisition functions themselves. Recent works have considered effective gradient-based approaches for acquisition optimization. Wilson et al. [77] demonstrates how stochastic first-order methods can be leveraged for optimizing Monte Carlo acquisition functions. Balandat et al. [6] build on this work and put forth sample average approximations for MC acquisition functions that admit gradient-based optimization using deterministic higher-order optimizers such as L-BFGS-B.

Another line of work proposes to switch from BO to local optimization based on some stopping criterion to achieve faster local convergence, using either zeroth order [60] or gradient-based [57] optimization. While McLeod et al. [57] are also concerned with numerical issues, we emphasize that those issues arise due to ill-conditioned covariance matrices and are orthogonal to the numerical pathologies of improvement-based acquisition functions.

## 3    Theoretical Analysis of Expected Improvement's Vanishing Gradients

In this section, we shed light on the conditions on the objective function and surrogate model that give rise to the numerically vanishing gradients in EI, as seen in Figure 1. In particular, we show that as a BO algorithm closes the optimality gap $f^* - y^*$, where $f^*$ is the global maximum of the function $f_{\text{true}}$, and the associated GP surrogate's uncertainty decreases, EI is exceedingly likely to exhibit numerically vanishing gradients.

Let $P_{\mathbf{x}}$ be a distribution over the inputs $\mathbf{x}$, and $f \sim P_f$ be an objective drawn from a Gaussian process. Then with high probability over the particular instantiation $f$ of the objective, the probability that an input $\mathbf{x} \sim P_{\mathbf{x}}$ gives rise to an argument $(\mu(\mathbf{x}) - y^*)/\sigma(\mathbf{x})$ to $h$ in Eq. (2) that is smaller than a threshold $B$ exceeds $P_{\mathbf{x}}(f(\mathbf{x}) < f^* - \epsilon_n)$, where $\epsilon_n$ depends on the optimality gap $f^* - y^*$ and the maximum posterior uncertainty $\max_{\mathbf{x}} \sigma_n(\mathbf{x})$. This pertains to EI's numerically vanishing values and gradients, since the numerical support $\mathcal{S}_\eta(h) = \{\mathbf{x} : |h(\mathbf{x})| > \eta\}$ of a naïve implementation of $h$ in (2) is limited by a lower bound $B(\eta)$ that depends on the floating point precision $\eta$. Formally, $\mathcal{S}_\eta(h) \subset [B(\eta), \infty)$ even though $\mathcal{S}_0(h) = \mathbb{R}$ mathematically. As a consequence, the following result can be seen as a bound on the probability of encountering numerically vanishing values and gradients in EI using samples from the distribution $P_{\mathbf{x}}$ to initialize the optimization of the acquisition function.

**Theorem 1.** *Suppose $f$ is drawn from a Gaussian process prior $P_f$, $y^* \leq f^*$, $\mu_n, \sigma_n$ are the mean and standard deviation of the posterior $P_f(f|\mathcal{D}_n)$ and $B \in \mathbb{R}$. Then with probability $1 - \delta$,*

$$P_{\mathbf{x}}\left(\frac{\mu_n(\mathbf{x}) - y^*}{\sigma_n(\mathbf{x})} < B\right) \geq P_{\mathbf{x}}\left(f(\mathbf{x}) < f^* - \epsilon_n\right) \tag{7}$$

*where $\epsilon_n = (f^* - y^*) + \left(\sqrt{-2\log(2\delta)} - B\right)\max_{\mathbf{x}} \sigma_n(\mathbf{x})$.*

For any given – and especially early – iteration, $\epsilon_n$ does not have to be small, as both the optimality gap and the maximal posterior standard deviation can be large initially. Note that under certain technical conditions on the kernel function and the asymptotic distribution of the training data $\mathcal{D}_n$, the maximum posterior variance vanishes guaranteeably as $n$ increases, see [50, Corollary 3.2]. On its own, Theorem 1 gives insight into the non-asymptotic behavior by exposing a dependence to the distribution of objective values $f$. In particular, if the set of inputs that give rise to high objective values ($\approx f^*$) is concentrated, $P(f(\mathbf{x}) < f^* - \epsilon)$ will decay very slowly as $\epsilon$ increases, thereby maintaining a lower bound on the probability of close to 1. As an example, this is the case for the Ackley function, especially as the dimensionality increases, which explains the behavior in Figure 1.

# 4 Unexpected Improvements

In this section, we propose re-formulations of analytic and MC-based improvement-based acquisition functions that render them significantly easier to optimize. We will use differing fonts, e.g. $\log$ and `log`, to differentiate between the mathematical functions and their numerical implementations.

## 4.1 Analytic LogEI

Mathematically, EI's values and gradients are nonzero on the entire real line, except in the noiseless case for points that are perfectly correlated with previous observations. However, naïve implementations of $h$ are *numerically* zero when $z = (\mu(\mathbf{x}) - y^*)/\sigma(\mathbf{x})$ is small, which happens when the model has high confidence that little improvement can be achieved at $\mathbf{x}$. We propose an implementation of $\log \circ h$ that can be accurately computed for a much larger range of inputs. Specifically, we compute

$$\text{LogEI}_{y^*}(\mathbf{x}) = \texttt{log\_h}((\mu(\mathbf{x}) - y^*)/\sigma(\mathbf{x})) + \texttt{log}(\sigma(\mathbf{x})), \tag{8}$$

where `log_h` is mathematically equivalent to $\log \circ h$ and can be stably and accurately computed by

$$\texttt{log\_h}(z) = \begin{cases} \texttt{log}(\phi(z) + z\Phi(z)) & z > -1 \\ -z^2/2 - c_1 + \texttt{log1mexp}(\log(\texttt{erfcx}(-z/\sqrt{2})|z|) + c_2) & -1/\sqrt{\epsilon} < z \le -1 \\ -z^2/2 - c_1 - 2\log(|z|) & z \le -1/\sqrt{\epsilon} \end{cases} \tag{9}$$

where $c_1 = \log(2\pi)/2$, and $c_2 = \log(\pi/2)/2$, $\epsilon$ is the numerical precision, and `log1mexp`, `erfcx` are numerically stable implementations of $\log(1 - \exp(z))$ and $\exp(z^2)\text{erfc}(z)$, respectively, see App. A. Progenitors of Eq. (9) are found in SMAC 1.0 [35] and RoBO [46] which contain a log-transformed analytic EI implementation that is much improved, but can still exhibit instabilities as $z$ grows negative, see App. Fig. 10. To remedy similar instabilities, we put forth the third, asymptotic case in Eq. (9), ensuring numerical stability throughout, see App. A.2 for details. The asymptotically quadratic behavior of `log_h` becomes apparent in the last two cases, making the function particularly amenable to gradient-based optimization with *significant* practical implications for EI-based BO.

## 4.2 Monte Carlo Parallel LogEI

Beyond analytic EI, Monte Carlo formulations of parallel EI that perform differentiation on the level of MC samples, don't just exhibit numerically, but mathematically zero gradients for a significant proportion of practically relevant inputs. For qEI, the primary issue is the discrete maximum over the $q$ outcomes for each MC sample in (5). In particular, the acquisition utility of expected improvement in Eq. 4 on a single sample $\xi_i$ of $f$ is $\max_j[\xi_i(\mathbf{x}_j) - y^*]_+$. Mathematically, we smoothly approximate the acquisition utility in two stages: 1) $u_{ij} = \text{softplus}_{\tau_0}(\xi_i(\mathbf{x}_j) - y^*) \approx [\xi_i(\mathbf{x}_j) - y^*]_+$ and 2) $\|u_i.\|_{1/\tau_{\max}} \approx \max_j u_{ij}$. Notably, while we use canonical softplus and p-norm approximations here, specialized fat-tailed non-linearities are required to scale to large batches, see Appendix A.4. Since the resulting quantities are strictly positive, they can be transformed to log-space permitting an implementation of qLogEI that is numerically stable and can be optimized effectively. In particular,

$$\begin{aligned} \text{qLogEI}_{y^*}(\mathbf{X}) &= \log \int \left( \sum_{j=1}^{q} \text{softplus}_{\tau_0}(f(\mathbf{x}_j) - y^*)^{1/\tau_{\max}} \right)^{\tau_{\max}} df \\ &\approx \texttt{logsumexp}_i(\tau_{\max}\texttt{logsumexp}_j(\texttt{logsoftplus}_{\tau_0}(\xi^i(\mathbf{x}_j) - y^*))/\tau_{\max})), \end{aligned} \tag{10}$$

where $i$ is the index of the Monte Carlo draws from the GP posterior, $j = 1, \ldots, q$ is the index for the candidate in the batch, and `logsoftplus` is a numerically stable implementation of $\log(\log(1 + \exp(z)))$. See Appendix A.3 for details, including the novel fat-tailed non-linearities like `fatplus`.

While the smoothing in (10) approximates the canonical qEI formulation, the following result shows that the associated relative approximation error can be quantified and bounded tightly as a function of the temperature parameters $\tau_0$, $\tau_{\max}$ and the batch size $q$. See Appendix C for the proof.

**Lemma 2.** *[Relative Approximation Guarantee] Given $\tau_0, \tau_{\max} > 0$, the approximation error of* qLogEI *to* qEI *is bounded by*

$$\left| e^{\text{qLogEI}(\mathbf{X})} - \text{qEI}(\mathbf{X}) \right| \le (q^{\tau_{\max}} - 1) \, \text{qEI}(\mathbf{X}) + \log(2)\tau_0 q^{\tau_{\max}}. \tag{11}$$

In Appendix D.10, we show the importance of setting the temperatures sufficiently low for qLogEI to achieve good optimization characteristics, something that only becomes possible by transforming all involved computations to log-space. Otherwise, the smooth approximation to the acquisition utility would exhibit vanishing gradients numerically, as the discrete $\max$ operator does mathematically.

### 4.3 Constrained EI

Both analytic and Monte Carlo variants of LogEI can be extended for optimization problems with black-box constraints. For analytic CEI with independent constraints of the form $f_i(\mathbf{x}) \leq 0$, the constrained formulation in Eq. (3) simplifies to $\mathrm{LogCEI}(\mathbf{x}) = \mathrm{LogEI}(\mathbf{x}) + \sum_i \log(P(f_i(\mathbf{x}) \leq 0))$, which can be readily and stably computed using LogEI in Eq. (8) and, if $f_i$ is modelled by a GP, a stable implementation of the Gaussian log cumulative distribution function. For the Monte Carlo variant, we apply a similar strategy as for Eq. (10) to the constraint indicators in Eq. (3): 1) a smooth approximation and 2) an accurate and stable implementation of its log value, see Appendix A.

### 4.4 Monte Carlo Parallel LogEHVI

The numerical difficulties of qEHVI in (6) are similar to those of qEI, and the basic ingredients of smoothing and log-transformations still apply, but the details are significantly more complex since qEHVI uses many operations that have mathematically zero gradients with respect to some of the inputs. Our implementation is based on the differentiable inclusion-exclusion formulation of the hypervolume improvement [13]. As a by-product, the implementation also readily allows for the differentiable computation of the expected log hypervolume, instead of the log expected hypervolume, note the order, which can be preferable in certain applications of multi-objective optimization [26].

## 5 Empirical Results

We compare standard versions of analytic EI (EI) and constrained EI (CEI), Monte Carlo parallel EI (qEI), as well as Monte Carlo EHVI (qEHVI), in addition to other state-of-the-art baselines like lower-bound Max-Value Entropy Search (GIBBON) [61] and single- and multi-objective Joint Entropy Search (JES) [36, 71]. All experiments are implemented using BoTorch [6] and utilize multi-start optimization of the AF with `scipy`'s L-BFGS-B optimizer. In order to avoid conflating the effect of BoTorch's default initialization strategy with those of our contributions, we use 16 initial points chosen uniformly at random from which to start the L-BFGS-B optimization. For a comparison with other initialization strategies, see Appendix D. We run multiple replicates and report mean and error bars of $\pm 2$ standard errors of the mean. Appendix D.1 contains additional details.

**Single-objective sequential BO**  We compare EI and LogEI on the 10-dimensional convex Sum-of-Squares (SoS) function $f(\mathbf{x}) = \sum_{i=1}^{10} (x_i - 0.5)^2$, using 20 restarts seeded from 1024 pseudo-random samples through BoTorch's default initialization heuristic. Figure 2 shows that due to vanishing gradients, EI is unable to make progress even on this trivial problem.

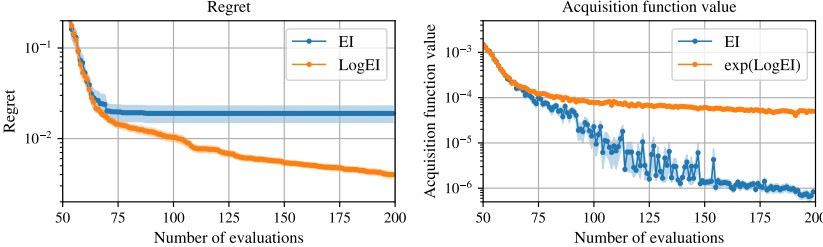

Figure 2: Regret and EI acquisition value for the candidates selected by maximizing EI and LogEI on the convex Sum-of-Squares problem. Optimization stalls out for EI after about 75 observations due to vanishing gradients (indicated by the jagged behavior of the acquisition value), while LogEI continues to make steady progress.

In Figure 3, we compare performance on the Ackley and Michalewicz test functions [67]. Notably, LogEI substantially outperforms EI on Ackley as the dimensionality increases. Ackley is a challenging multimodal function for which it is critical to trade off local exploitation with global exploration, a task made exceedingly difficult by the numerically vanishing gradients of EI in a large fraction of the search space. We see a similar albeit less pronounced behavior on Michalewicz, which reflects the fact that Michalewicz is a somewhat less challenging problem than Ackley.

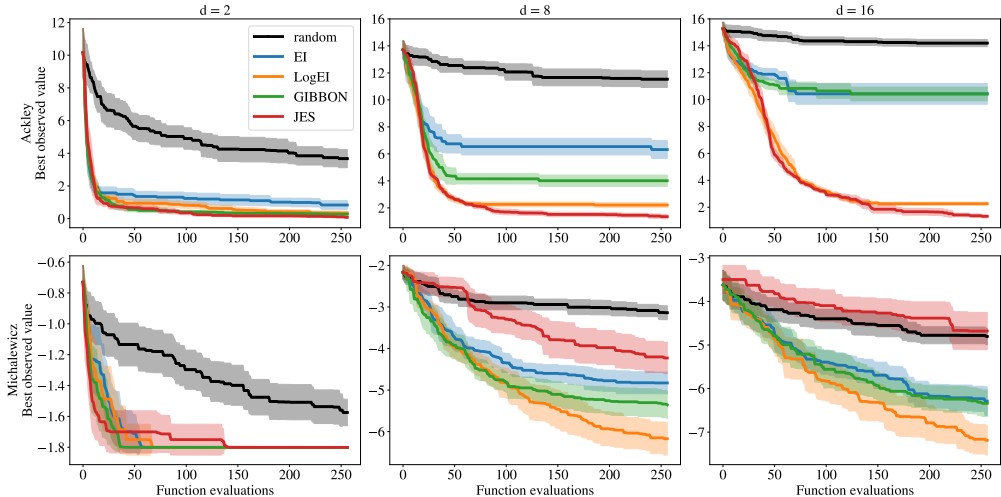

Figure 3: Best objective value as a function of iterations on the moderately and severely non-convex Michalewicz and Ackley problems for varying numbers of input dimensions. LogEI substantially outperforms both EI and GIBBON, and this gap widens as the problem dimensionality increases. JES performs slightly better than LogEI on Ackley, but for some reason fails on Michalewicz. Notably, JES is almost two orders of magnitude slower than the other acquisition functions (see Appendix D).

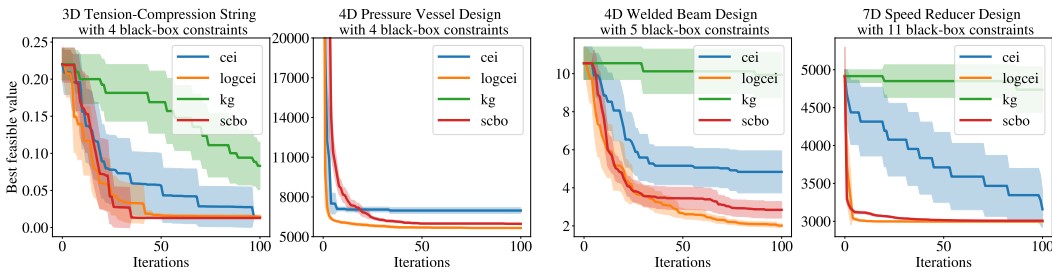

Figure 4: Best feasible objective value as a function of number of function evaluations (iterations) on four engineering design problems with black-box constraints after an initial $2d$ pseudo-random evaluations.

**BO with Black Box Constraints**    Figure 4 shows results on four engineering design problems with black box constraints that were also considered in [22]. We apply the same bilog transform as the trust region-based SCBO method [22] to all constraints to make them easier to model with a GP. We see that LogCEI outperforms the naive CEI implementation and converges faster than SCBO. Similar to the unconstrained problems, the performance gains of LogCEI over CEI grow with increasing problem dimensionality and the number of constraints. Notably, we found that for some problems, LogCEI in fact *improved upon some of the best results quoted in the original literature*, while using three orders of magnitude fewer function evaluations, see Appendix D.7 for details.

**Parallel Expected Improvement with qLogEI**    Figure 5 reports the optimization performance of parallel BO on the 16-dimensional Ackley function for both sequential greedy and joint batch optimization using the fat-tailed non-linearities of App. A.4. In addition to the apparent advantages of qLogEI over qEI, a key finding is that jointly optimizing the candidates of batch acquisition functions can yield highly competitive optimization performance, see App. D.3 for extended results.

**High-dimensional BO with qLogEI**    Figure 6 shows the performance of LogEI on three high-dimensional problems: the 6-dimensional Hartmann function embedded in a 100-dimensional space, a 100-dimensional rover trajectory planning problem, and a 103-dimensional SVM hyperparameter tuning problem. We use a 103-dimensional version of the 388-dimensional SVM problem considered by Eriksson and Jankowiak [21], where the 100 most important features were selected using Xgboost.

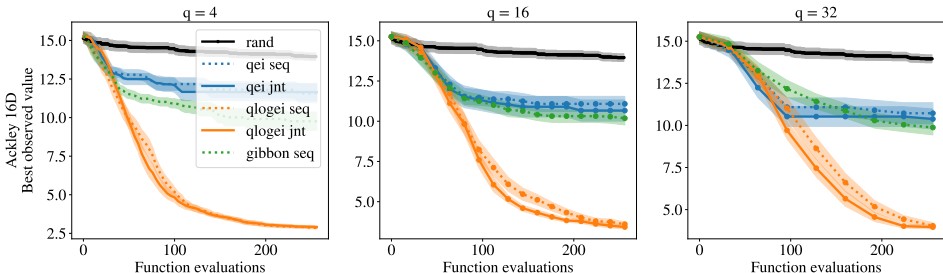

Figure 5: Best objective value for parallel BO as a function of the number evaluations for single-objective optimization on the 16-dimensional Ackley function with varying batch sizes $q$. Notably, joint optimization of the batch outperforms sequential greedy optimization.

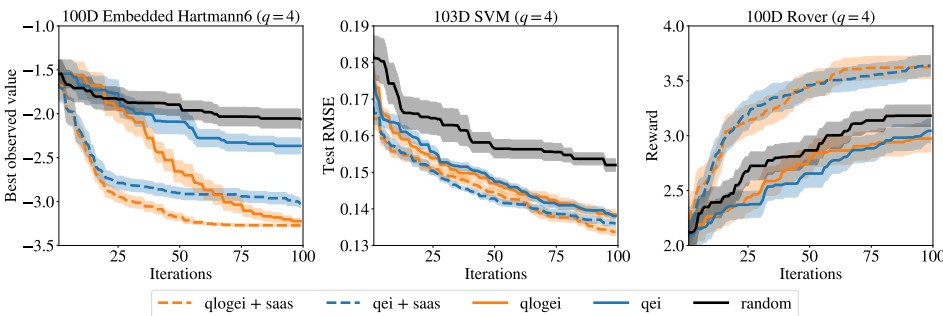

Figure 6: Best objective value as a function of number of function evaluations (iterations) on three high-dimensional problems, including Eriksson and Jankowiak [21]'s SAAS prior.

Figure 6 shows that the optimization exhibits varying degrees of improvement from the inclusion of qLogEI, both when combined with SAASBO [21] and a standard GP. In particular, qLogEI leads to significant improvements on the embedded Hartmann problem, even leading BO with the canonical GP to ultimately catch up with the SAAS-prior-equipped model. On the other hand, the differences on the SVM and Rover problems are not significant, see Section 6 for a discussion.

**Multi-Objective optimization with qLogEHVI**  Figure 7 compares qLogEHVI and qEHVI on two multi-objective test problems with varying batch sizes, including the real-world-inspired cell network design for optimizing coverage and capacity [19]. The results are consistent with our findings in the single-objective and constrained cases: qLogEHVI consistently outperforms qEHVI and even JES [71] for all batch sizes. Curiously, for the largest batch size and DTLZ2, qLogNEHVI's improvement over the reference point (HV > 0) occurs around three batches after the other methods, but dominates their performance in later batches. See Appendix D.5 for results on additional synthetic and real-world-inspired multi-objective problems such as the laser plasma acceleration optimization [38], and vehicle design optimization [54, 68]

## 6    Discussion

To recap, EI exhibits vanishing gradients 1) when high objective values are highly concentrated in the search space, and 2) as the optimization progresses. In this section, we highlight that these conditions are not met for all BO applications, and that LogEI's performance depends on the surrogate's quality.

**On problem dimensionality**  While our experimental results show that advantages of LogEI generally grow larger as the dimensionality of the problem grows, we stress that this is fundamentally due to the concentration of high objective values in the search space, not the dimensionality itself. Indeed, we have observed problems with high ambient dimensionality but low intrinsic dimensionality, where LogEI does not lead to significant improvements over EI, e.g. the SVM problem in Figure 6.

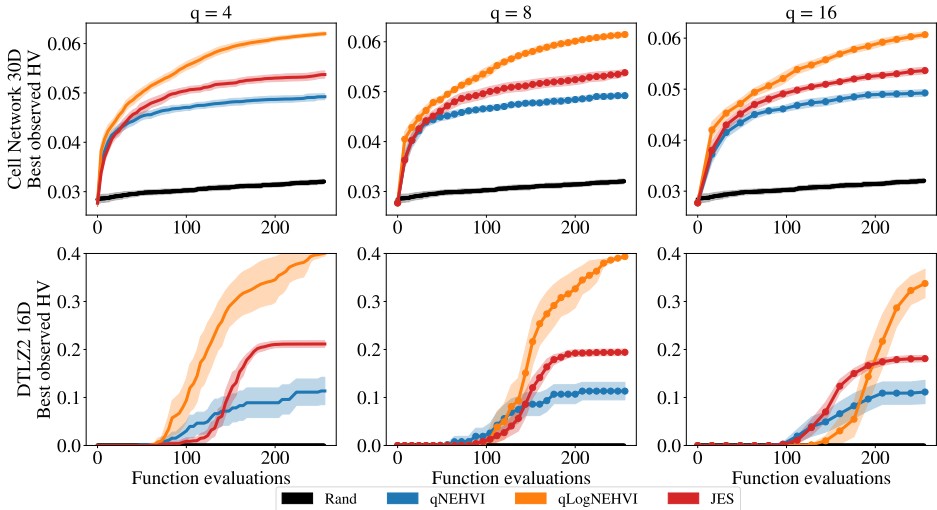

Figure 7: Batch optimization performance on two multi-objective problems, as measured by the hypervolume of the Pareto frontier across observed points. This plot includes JES [71]. Similar to the single-objective case, the LogEI variant qLogEHVI significantly outperforms the baselines.

**On asymptotic improvements** While members of the LogEI family can generally be optimized better, leading to higher acquisition values, improvements in optimization performance might be small in magnitude, e.g. the log-objective results on the convex 10D sum of squares in Fig. 2, or only begin to materialize in later iterations, like for $q = 16$ on DTLZ2 in Figure 7.

**On model quality** Even if good objective values are concentrated in a small volume of the search space and many iterations are run, LogEI might still not outperform EI if the surrogate's predictions are poor, or its uncertainties are not indicative of the surrogate's mismatch to the objective, see Rover in Fig. 6. In these cases, better acquisition values do not necessarily lead to better BO performance.

**Replacing EI** Despite these limitation, we strongly suggest replacing variants of EI with their LogEI counterparts. If LogEI were dominated by EI on some problem, it would be an indication that the EI family itself is sub-optimal, and improvements in performance can be attributed to the exploratory quality of randomly distributed candidates, which could be incorporated explicitly.

## 7 Conclusion

Our results demonstrate that the problem of vanishing gradients is a major source of the difficulty of optimizing improvement-based acquisition functions and that we can mitigate this issue through careful reformulations and implementations. As a result, we see substantially improved optimization performance across a variety of modified EI variants across a broad range of problems. In particular, we demonstrate that joint batch optimization for parallel BO can be competitive with, and at times exceed the sequential greedy approach typically used in practice, which also benefits from our modifications. Besides the convincing performance improvements, one of the key advantages of our modified acquisition functions is that they are much less dependent on heuristic and potentially brittle initialization strategies. Moreover, our proposed modifications do not meaningfully increase the computational complexity of the respective original acquisition function.

While our contributions may not apply verbatim to other classes of acquisition functions, our key insights and strategies do translate and could help with e.g. improving information-based [34, 76], cost-aware [51, 66], and other types of acquisition functions that are prone to similar numerical challenges. Further, combining the proposed methods with gradient-aware first-order BO methods [5, 16, 23] could lead to particularly effective high-dimensional applications of BO, since the advantages of both methods tend to increase with the dimensionality of the search space. Overall, we hope that our findings will increase awareness in the community for the importance of optimizing acquisition functions well, and in particular, for the care that the involved numerics demand.

## Acknowledgments and Disclosure of Funding

The authors thank Frank Hutter for valuable references about prior work on numerically stable computations of analytic EI, David Bindel for insightful conversations about the difficulty of optimizing EI, as well as the anonymous reviewers for their knowledgeable feedback.

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

## A  Acquisition Function Details

### A.1  Analytic Expected Improvement

Recall that the main challenge with computing analytic LogEI is to accurately compute $\log h$, where $h(z) = \phi(z) + z\Phi(z)$, with $\phi(z) = \exp(-z^2/2)/\sqrt{2\pi}$ and $\Phi(z) = \int_{-\infty}^{z} \phi(u)du$. To express $\log h$ in a numerically stable form as $z$ becomes increasingly negative, we first take the log and multiply $\phi$ out of the argument to the logarithm:

$$\log h(z) = z^2/2 - \log(2\pi)/2 + \log\left(1 + z\frac{\Phi(z)}{\phi(z)}\right). \tag{12}$$

Fortunately, this form exposes the quadratic factor, $\Phi(z)/\phi(z)$ can be computed via standard implementations of the scaled complementary error function $\texttt{erfcx}$, and $\log(1 + z\Phi(z)/\phi(z))$, the last term of Eq. (12), can be computed stably with the $\texttt{log1mexp}$ implementation proposed in [56]:

$$\texttt{log1mexp}(x) = \begin{cases} \texttt{log}(-\texttt{expm1}(x)) & -\log 2 < x \\ \texttt{log1p}(-\texttt{exp}(x)) & -\log 2 \geq x \end{cases}, \tag{13}$$

where $\texttt{expm1}$, $\texttt{log1p}$ are canonical stable implementations of $\exp(x) - 1$ and $\log(1+x)$, respectively. In particular, we compute

$$\texttt{log\_h}(z) = \begin{cases} \texttt{log}(\phi(z) + z\Phi(z)) & z > -1 \\ -z^2/2 - c_1 + \texttt{log1mexp}(\log(\texttt{erfcx}(-z/\sqrt{2})|z|) + c_2) & 1/\sqrt{\epsilon} < z \leq -1 \\ -z^2/2 - c_1 - 2\log(|z|) & z \leq -1/\sqrt{\epsilon}. \end{cases} \tag{14}$$

where $c_1 = \log(2\pi)/2$, $c_2 = \log(\pi/2)/2$, and $\epsilon$ is the floating point precision. We detail the reasoning behind the third asymptotic case of Equation (14) in Lemma 4 of Section A.2 below.

**Numerical Study of Acquisition Values**  Figure 8 shows both the numerical failure mode of a naïve implementation of EI, which becomes *exactly* zero numerically for moderately small $z$, while the evaluation via $\texttt{log\_h}$ in Eq. (14) exhibits quadratic asymptotic behavior that is particularly amenable to numerical optimization routines.

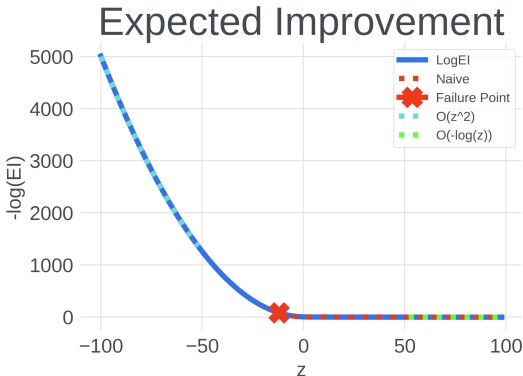

Figure 8:  Plot of the $\log h$, computed via $\log \circ h$ and $\texttt{log\_h}$ in Eq. (14). Crucially, the naïve implementation fails as $z = (\mu(\mathbf{x}) - f^*)/\sigma(\mathbf{x})$ becomes increasingly negative, due to being exactly numerically zero, while our proposed implementation exhibits quadratic asymptotic behavior.

**Equivalence of Optimizers**  Lemma 3 states that if the maximum of EI is greater than 0, LogEI and EI have the same set of maximizers. Furthermore, if $\max_{\mathbf{x}\in\mathbb{X}} \text{EI}(\mathbf{x}) = 0$, then $\mathbb{X} = \arg\max_{\mathbf{x}\in\mathbb{X}} \text{EI}(\mathbf{x})$. In this case, LogEI is undefined everywhere, so it has no maximizers, which we note would yield the same BO policy as EI, for which every point is a maximizer.

**Lemma 3.**  *If* $\max_{\mathbf{x}\in\mathbb{X}} \text{EI}(\mathbf{x}) > 0$*, then* $\arg\max_{\mathbf{x}\in\mathbb{X}} \text{EI}(\mathbf{x}) = \arg\max_{\mathbf{x}\in\mathbb{X}, \text{EI}(\mathbf{x})>0} \text{LogEI}(\mathbf{x})$.

*Proof.*  Suppose $\max_{\mathbf{x}\in\mathbb{X}} \text{EI}(\mathbf{x}) > 0$. Then $\arg\max_{\mathbf{x}\in\mathbb{X}} \text{EI}(\mathbf{x}) = \arg\max_{\mathbf{x}\in\mathbb{X}, \text{EI}(\mathbf{x})>0} \text{EI}(\mathbf{x})$. For all $\mathbf{x} \in \mathbb{X}$ such that $\text{EI}(\mathbf{x}) > 0$, $\text{LogEI}(\mathbf{x}) = \log(\text{EI}(\mathbf{x}))$. Since $\log$ is monotonic, we have that $\arg\max_{z\in\mathbb{R}_{>0}} z = \arg\max_{z\in\mathbb{R}_{>0}} \log(z)$. Hence, $\arg\max_{\mathbf{x}\in\mathbb{X}, \text{EI}(\mathbf{x})>0} \text{EI}(\mathbf{x}) = \arg\max_{\mathbf{x}\in\mathbb{X}, \text{EI}(\mathbf{x})>0} \text{LogEI}(\mathbf{x})$. $\qquad\square$

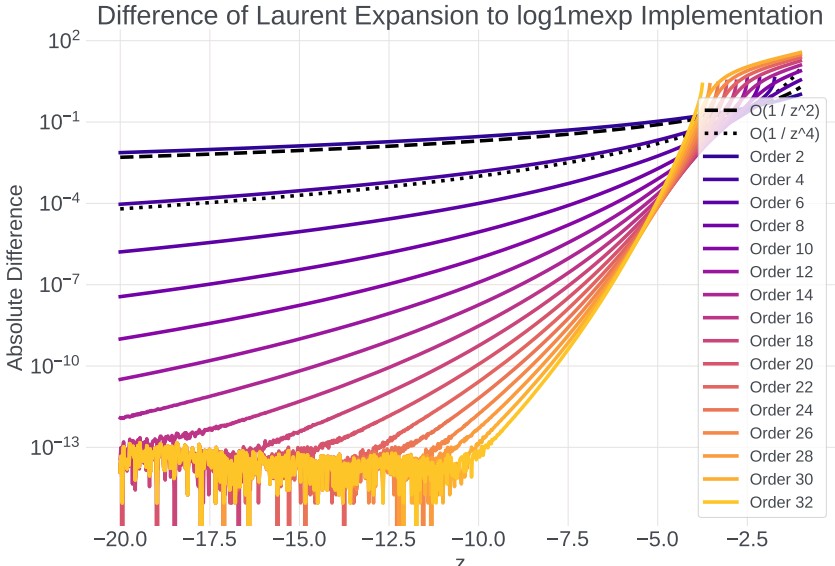

Figure 9: Convergence behavior of the asymptotic Laurent expansion Eq. (16) of different orders.

## A.2 Analytical LogEI's Asymptotics

As $z$ grows negative and large, even the more robust second branch in Eq. (14), as well as the implementation of Hutter et al. [35] and Klein et al. [46] can suffer from numerical instabilities (Fig. 10, left). In our case, the computation of the last term of Eq. (12) is problematic for large negative $z$. For this reason, we propose an approximate asymptotic computation based on a Laurent expansion at $-\infty$. As a result of the full analysis in the following, we also attain a particularly simple formula with inverse quadratic convergence in $z$, which is the basis of the third branch of Eq. (14):

$$\log\left(1 + \frac{z\Phi(z)}{\phi(z)}\right) = -2\log(|z|) + \mathcal{O}(|z^{-2}|). \tag{15}$$

In full generality, the asymptotic behavior of the last term can be characterized by the following result.

**Lemma 4** (Asymptotic Expansion)**.** *Let $z < -1$ and $K \in \mathbb{N}$, then*

$$\log\left(1 + \frac{z\Phi(z)}{\phi(z)}\right) = \log\left(\sum_{k=1}^{K}(-1)^{k+1}\left[\prod_{j=0}^{k-1}(2j+1)\right]z^{-2k}\right) + \mathcal{O}(|z^{-2(K-1)}|). \tag{16}$$

*Proof.* We first derived a Laurent expansion of the non-log-transformed $z\Phi(z)/\phi(z)$, a key quantity in the last term of Eq. (12), with the help of Wolfram Alpha [37]:

$$\frac{z\Phi(z)}{\phi(z)} = -1 - \sum_{k=1}^{K}(-1)^{k}\left[\prod_{j=0}^{k-1}(2j+1)\right]z^{-2k} + \mathcal{O}(|z|^{-2K}). \tag{17}$$

It remains to derive the asymptotic error bound through the log-transformation of the above expansion. Letting $L(z,K) = \sum_{k=1}^{K}(-1)^{k+1}\left[\prod_{j=0}^{k-1}(2j+1)\right]z^{-2k}$, we get

$$\begin{aligned}
\log\left(1 + \frac{z\Phi(z)}{\phi(z)}\right) &= \log\left(L(z,K) + \mathcal{O}(|z|^{-2K})\right) \\
&= \log L(z,K) + \log(1 + \mathcal{O}(|z|^{-2K})/L(z,K)) \\
&= \log L(z,K) + \mathcal{O}(\mathcal{O}(|z|^{-2K})/L(z,K)) \\
&= \log L(z,K) + \mathcal{O}(|z|^{-2(K-1)}).
\end{aligned} \tag{18}$$

The penultimate equality is due to $\log(1+x) = x + \mathcal{O}(x^2)$, the last due to $L(z,K) = \Theta(|z|^{-2})$. $\quad\square$

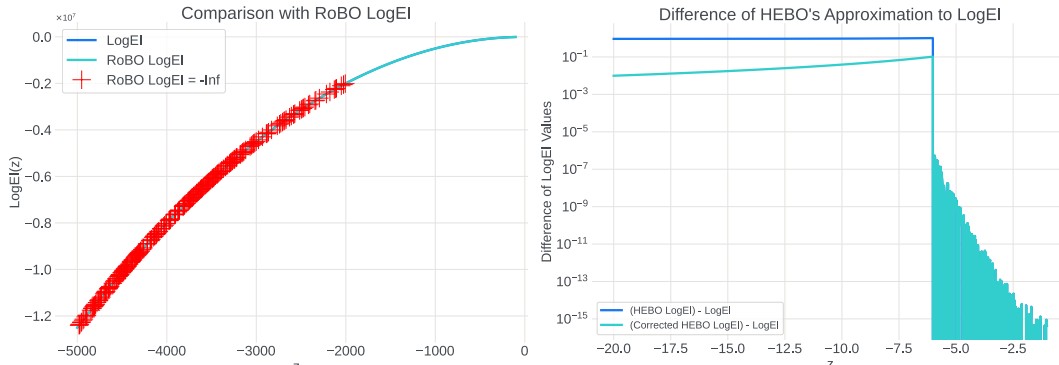

Figure 10: Left: Comparison of LogEI values in single-precision floating point arithmetic, as a function of $z = (\mu(x) - f^*)/\sigma(x)$ to RoBO's LogEI implementation [46]. Notably, RoBO's LogEI improves greatly on the naïve implementation (Fig. 8), but still exhibits failure points (red) well above floating point underflow. The implementation of Eq. (14) continues to be stable in this regime. Right: Comparison of LogEI values to HEBO's approximate LogEI implementation [12]. At the time of writing, HEBO's implementation exhibits a discontinuity and error of $> 1.02$ at the threshold $z = -6$, below which the approximation takes effect. The discontinuity could be ameliorated, though not removed, by correcting the log normalization constant (turquoise). The figure also shows that the naïve implementation used by HEBO for $z > -6$ starts to become unstable well before $z = -6$.

**SMAC 1.0 and RoBO's Analytic LogEI**  To our knowledge, SMAC 1.0's implementation of the logarithm of analytic EI due to Hutter et al. [35], later translated to RoBO [46], was the first to improve the numerical stability of analytic EI through careful numerics. The associated implementation is mathematically identical to $\log \circ$ EI, and greatly improves the numerical stability of the computation. For large negative $z$ however, the implementation still exhibits instabilities that gives rise to floating point infinities through which useful gradients cannot be propagated (Fig. 10, left). The implementation proposed herein remedies this problem by switching to the asymptotic approximation of Eq. (15) once it is accurate to machine precision $\epsilon$. This is similar to the use of asymptotic expansions for the computation of $\alpha$-stable densities proposed by Ament and O'Neil [2].

**HEBO's Approximate Analytic LogEI**  HEBO [12] contains an approximation to the logarithm of analytical EI as part of its implementation of the MACE acquisition function [55], which – at the time of writing – is missing the log normalization constant of the Gaussian density, leading to a large discontinuity at the chosen cut-off point of $z = -6$ below which the approximation takes effect, see here for the HEBO implementation. Notably, HEBO does not implement an *exact* stable computation of LogEI like the one of Hutter et al. [35], Klein et al. [46] and the non-asymptotic branches of the current work. Instead, it applies the approximation for all $z < -6$, where the implementation exhibits a maximum error of $> 1.02$, or if the implementation's normalization constant were corrected, a maximum error of $> 0.1$. By comparison, the implementation put forth herein is mathematically exact for the non-asymptotic regime $z > -1/\sqrt{\epsilon}$ and accurate to numerical precision in the asymptotic regime due to the design of the threshold value.

### A.3   Monte-Carlo Expected Improvement

For Monte-Carlo, we cannot directly apply similar numerical improvements as for the analytical version, because the utility values, the integrand of Eq. (4), on the sample level are likely to be *mathematically* zero. For this reason, we first smoothly approximate the acquisition utility and subsequently apply log transformations to the approximate acquisition function.

To this end, a natural choice is $\mathrm{softplus}_{\tau_0}(x) = \tau_0 \log(1 + \exp(x/\tau_0))$ for smoothing the $\max(0, x)$, where $\tau_0$ is a temperature parameter governing the approximation error. Further, we approximate the $\max_i$ over the $q$ candidates by the norm $\| \cdot \|_{1/\tau_{\max}}$ and note that the approximation error introduced by both smooth approximations can be bound tightly as a function of two "temperature" parameters $\tau_0$ and $\tau_{\max}$, see Lemma 2.

Importantly, the smoothing alone only solves the problem of having mathematically zero gradients, not that of having numerically vanishing gradients, as we have shown for the analytical case above. For this reason, we transform all smoothed computations to log space and thus need the following special implementation of $\log \circ \mathtt{softplus}$ that can be evaluated stably for a very large range of inputs:

$$
\mathtt{logsoftplus}_\tau(x) = \begin{cases} [\log \circ \mathtt{softplus}_\tau](x) & x/\tau > l \\ x/\tau + \log(\tau) & x/\tau \le l \end{cases}
$$

where $\tau$ is a temperature parameter and $l$ depends on the floating point precision used, around $-35$ for double precision in our implementation.

Note that the lower branch of $\mathtt{logsoftplus}$ is approximate. Using a Taylor expansion of $\log(1+z) = z - z^2/2 + \mathcal{O}(z^3)$ around $z = 0$, we can see that the approximation error is $\mathcal{O}(z^2)$, and therefore, $\log(\log(1 + \exp(x))) = x + \mathcal{O}(\exp(x)^2)$, which converges to $x$ exponentially quickly as $x \to -\infty$. In our implementation, $l$ is chosen so that no significant digit is lost in dropping the second order term from the lower branch.

Having defined $\mathtt{logsoftplus}$, we further note that

$$
\log \|\mathbf{x}\|_{1/\tau_{\max}} = \log \left( \sum_i x_i^{1/\tau_{\max}} \right)^{\tau_{\max}}
$$

$$
= \tau_{\max} \log \left( \sum_i \exp(\log(x_i)/\tau_{\max}) \right)
$$

$$
= \tau_{\max} \mathtt{logsumexp}_i \left( \log(x_i)/\tau_{\max} \right)
$$

Therefore, we express the logarithm of the smoothed acquisition utility for $q$ candidates as

$$
\tau_{\max} \mathtt{logsumexp}_j^q (\mathtt{logsoftplus}_{\tau_0}((\xi^i(x_j) - y^*)/\tau_{\max}).
$$

Applying another $\mathtt{logsumexp}$ to compute the logarithm of the mean of acquisition utilities over a set of Monte Carlo samples $\{\xi_i\}_i$ gives rise to the expression in Eq. (10).

In particular for large batches (large $q$), this expression can still give rise to vanishing gradients for some candidates, which is due to the large dynamic range of the outputs of the $\mathtt{logsoftplus}$ when $x << 0$. To solve this problem, we propose a new class of smooth approximations to the "hard" non-linearities that decay as $\mathcal{O}(1/x^2)$ as $x \to -\infty$ in the next section.

### A.4    A Class of Smooth Approximations with Fat Tails for Larger Batches

A regular $\mathrm{softplus}(x) = \log(1 + \exp(x))$ function smoothly approximates the ReLU non-linearity and – in conjunction with the log transformations – is sufficient to achieve good numerical behavior for small batches of the Monte Carlo acquisition functions. However, as more candidates are added, $\log \mathrm{softplus}(x) = \log(\log(1 + \exp(x)))$ is increasingly likely to have a high dynamic range as for $x \ll 0$, $\log \mathrm{softplus}_\tau(x) \sim -x/\tau$. If $\tau > 0$ is chosen to be small, $(-x/\tau)$ can vary orders of magnitude within a single batch. This becomes problematic when we approximate the maximum utility over the batch of candidates, since $\mathtt{logsumexp}$ only propagates numerically non-zero gradients to inputs that are no smaller than approximately $(\max_j x_j - 700)$ in double precision, another source of vanishing gradients.

To solve this problem, we propose a new smooth approximation to the ReLU, maximum, and indicator functions that decay only polynomially as $x \to -\infty$, instead of exponentially, like the canonical softplus. The high level idea is to use $(1 + x^2)^{-1}$, which is proportional to the Cauchy density function (and is also known as a Lorentzian), in ways that maintain key properties of existing smooth approximations – convexity, positivity, etc – while changing the asymptotic behavior of the functions from exponential to $\mathcal{O}(1/x^2)$ as $x \to -\infty$, also known as a "fat tail". Further, we will show that the proposed smooth approximations satisfy similar maximum error bounds as their exponentially decaying counterparts, thereby permitting a similar approximation guarantee as Lemma 2 with minor adjustments to the involved constants. While the derivations herein are based on the Cauchy density with inverse quadratic decay, it is possible to generalize the derivations to e.g. $\alpha$-stable distribution whose symmetric variants permit accurate and efficient numerical computation [2].

**Fat Softplus**   We define

$$\varphi_+(x) = \alpha(1 + x^2)^{-1} + \log(1 + \exp(x)), \tag{19}$$

for a positive scalar $\alpha$. The following result shows that we can ensure the monotonicity and convexity – both important properties of the ReLU that we would like to maintain in our approximation – of $g$ by carefully choosing $\alpha$.

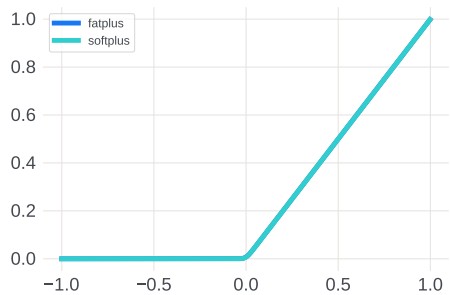
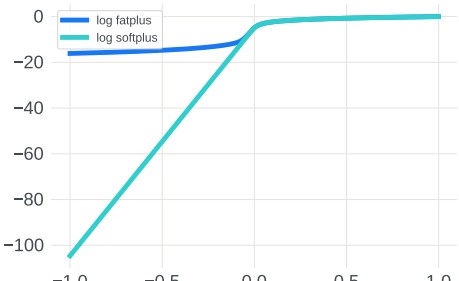

Figure 11:   The fat softplus approximates $\max(x, 0)$ similarly tightly as the regular softplus and is also monotonic, convex, and positive. The plot used a temperature of $\tau_0 = 0.01$.

Figure 12: The fat softplus has an $\mathcal{O}(1/x^2)$ asymptotic decay, versus the $\mathcal{O}(\exp(x))$ decay as $x \to -\infty$, moderating the dynamic range of the quantities involved in parallel LogEI.

**Lemma 5** (Monotonicity and Convexity). *$\varphi_+(x)$ is positive, monotonically increasing, and strictly convex for $\alpha$ satisfying*

$$0 \le \alpha < \frac{e^{1/\sqrt{3}}}{2\left(1 + e^{1/\sqrt{3}}\right)}.$$

*Proof.*  Positivity follows due to $\alpha \ge 0$, and both sumands being positive. Monotonicity and convexity can be shown via canonical differential calculus and bounding relevant quantities.

In particular, regarding monotonicity, we want to select $\alpha$ so that the first derivative is bounded below by zero:

$$\partial_x \varphi_+(x) = \frac{e^x}{1 + e^x} - \alpha \frac{2x}{(1 + x^2)^2}$$

First, we note that $\partial_x \varphi_+(x)$ is positive for $x < 0$ and any $\alpha$, since both terms are positive in this regime. For $x \ge 0$, $\frac{e^x}{1+e^x} = (1 + e^{-x})^{-1} \ge 1/2$, and $-1/(1 + x^2)^2 \ge -1/(1 + x^2)$, so that

$$\partial_x \varphi_+(x) \ge \frac{1}{2} - \alpha \frac{2x}{(1 + x^2)}$$

Forcing $\frac{1}{2} - \alpha \frac{2x}{(1+x^2)} > 0$, and multiplying by $(1 + x^2)$ gives rise to a quadratic equation whose roots are $x = 2\alpha \pm \sqrt{4\alpha^2 - 1}$. Thus, there are no real roots for $\alpha < 1/2$. Since the derivative is certainly positive for the negative reals and the guaranteed non-existence of roots implies that the derivative cannot cross zero elsewhere, $0 \le \alpha < 1/2$ is a sufficient condition for monotonicity of $\varphi_+$.

Regarding convexity, our goal is to prove a similar condition on $\alpha$ that guarantees the positivity of the second derivative:

$$\partial_x^2 \varphi_+(x) = \alpha \frac{6x^2 - 2}{(1 + x^2)^3} + \frac{e^{-x}}{(1 + e^{-x})^2}$$

Note that $\frac{6x^2 - 2}{(1+x^2)^3}$ is symmetric around 0, is negative in $(-\sqrt{1/3}, \sqrt{1/3})$ and has a minimum of $-2$ at 0. $\frac{e^{-x}}{(1+e^{-x})^2}$ is symmetric around zero and decreasing away from zero. Since the rational polynomial is only negative in $(-\sqrt{1/3}, \sqrt{1/3})$, we can lower bound $\frac{e^{-x}}{(1+e^{-x})^2} > \frac{e^{-\sqrt{1/3}}}{(1+e^{-\sqrt{1/3}})^2}$ in

$(-\sqrt{1/3}, \sqrt{1/3})$. Therefore,

$$\partial_x^2 \varphi_+(x) \geq \frac{e^{-x}}{(1 + e^{-x})^2} - 2\alpha$$

Forcing $\frac{e^{-\sqrt{1/3}}}{(1+e^{-\sqrt{1/3}})^2} - 2\alpha > 0$ and rearranging yields the result. Since $\frac{e^{-\sqrt{1/3}}}{(1+e^{-\sqrt{1/3}})^2}/2 \sim 0.115135$, the convexity condition is stronger than the monotonicity condition and therefore subsumes it. $\quad\square$

Importantly $\varphi$ decays only polynomially for increasingly negative inputs, and therefore $\log \varphi$ only logarithmically, which keeps the range of $\varphi$ constrained to values that are more manageable numerically. Similar to Lemma 7, one can show that

$$|\tau\varphi_+(x/\tau) - \mathrm{ReLU}(x)| \leq (\alpha + \log(2))\,\tau. \tag{20}$$

There are a large number of approximations or variants of the ReLU that have been proposed as activation functions of artificial neural networks, but to our knowledge, none satisfy the properties that we seek here: (1) smoothness, (2) positivity, (3) monotonicity, (4) convexity, and (5) polynomial decay. For example, the leaky ReLU does not satisfy (1) and (2), and the ELU does not satisfy (5).

**Fat Maximum** The canonical `logsumexp` approximation to $\max_i x_i$ suffers from numerically vanishing gradients if $\max_i x_i - \min_j x_j$ is larger a moderate threshold, around 760 in double precision, depending on the floating point implementation. In particular, while elements close to the maximum receive numerically non-zero gradients, elements far away are increasingly likely to have a numerically zero gradient. To fix this behavior for the smooth maximum approximation, we propose

$$\varphi_{\max}(\mathbf{x}) = \max_j x_j + \tau \log \sum_i \left[ 1 + \left( \frac{x_i - \max_j x_j}{\tau} \right)^2 \right]^{-1}. \tag{21}$$

This approximation to the maximum has the same error bound to the true maximum as the `logsumexp` approximation:

**Lemma 6.** *Given $\tau > 0$*

$$\max_i x_i \leq \tau\,\phi_{\max}(x/\tau) \leq \max_i x_i + \tau \log(d). \tag{22}$$

*Proof.* Regarding the lower bound, let $i = \arg\max_j x_j$. For this index, the associated summand in (21) is 1. Since all sumands are positive, the entire sum is lower bounded by 1, hence

$$\tau \log \sum_i \left[ 1 + \left( \frac{x_i - \max_j x_j}{\tau} \right)^2 \right]^{-1} > \tau \log(1) = 0$$

Adding $\max_j x_j$ to the inequality finishes the proof for the lower bound.

Regarding the upper bound, (21) can be maximized when $x_i = \max_j x_j$ for all $i$, in which case each $(x_i - \max_j x_j)^2$ is minimized, and hence each summand is maximized. In this case,

$$\tau \log \sum_i \left[ 1 + \left( \frac{x_i - \max_j x_j}{\tau} \right)^2 \right]^{-1} \leq \tau \log \left( \sum_i 1 \right) = \tau \log(d).$$

Adding $\max_j x_j$ to the inequality finishes the proof for the upper bound. $\quad\square$

**Fat Sigmoid** Notably, we encountered a similar problem using regular (log)-sigmoids to smooth the constraint indicators for EI with black-box constraints. In principle the Cauchy cummulative distribution function would satisfy these conditions, but requires the computation of $\arctan$, a special function that requires more floating point operations to compute numerically than the following function. Here, we want the smooth approximation $\iota$ to satisfy 1) positivity, 2) monotonicity, 3) polynomial decay, and 4) $\iota(x) = 1/2 - \iota(-x)$. Let $\gamma = \sqrt{1/3}$, then we define

$$\iota(x) = \begin{cases} \frac{2}{3} \left( 1 + (x - \gamma)^2 \right)^{-1} & x < 0, \\ 1 - \frac{2}{3} \left( 1 + (x + \gamma)^2 \right)^{-1} & x \geq 0. \end{cases}$$

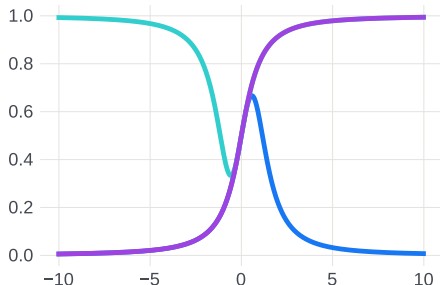 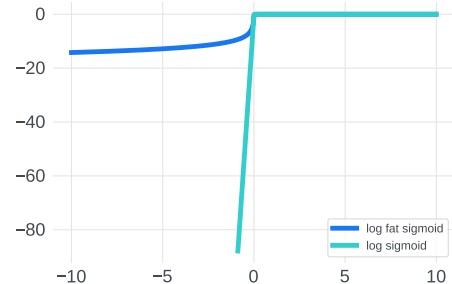

Figure 13: We construct the fat sigmoid approximation (purple) by splicing together two Lorentzians (blue and teal) at one of their inflection points.

Figure 14: The fat sigmoid approximation ($\tau = 0.01$) decays as $\mathcal{O}(1/x^2)$ instead of $\mathcal{O}(\exp(x))$ as $x \to -\infty$, minimizing the dynamic range of the numerical quantities in constrained qLogEI.

$\iota$ is monotonically increasing, satisfies $\iota(x) \to 1$ as $x \to \infty$, $\iota(0) = 1/2$, and $\iota(x) = \mathcal{O}(1/x^2)$ as $x \to -\infty$. Further, we note that the asymptotics are primarily important here, but that we can also make the approximation tighter by introducing a temperature parameter $\tau$, and letting $\iota_\tau(x) = \iota(x/\tau)$. The approximation error of $\iota_\tau(x)$ to the Heaviside step function becomes tighter point-wise as $\tau \to 0+$, except for at the origin where $\iota_\tau(x) = 1/2$, similar to the canonical sigmoid.

### A.5 Constrained Expected Improvement

For the analytical case, many computational frameworks already provide a numerically stable implementation of the logarithm of the Gaussian cummulative distribution function, in the case of PyTorch, `torch.special.log_ndtr`, which can be readily used in conjunction with our implementation of LogEI, as described in Sec. 4.3.

For the case of Monte-Carlo parallel EI, we implemented the fat-tailed $\iota$ function from Sec. A.4 to approximate the constraint indicator and compute the per-candidate, per-sample acquisition utility using

$$\left(\texttt{logsoftplus}_{\tau_0}(\xi_i(\mathbf{x}_j) - y^*) + \sum_k \log \circ \iota \left(-\frac{\xi_i^{(k)}(\mathbf{x}_j)}{\tau_{\text{cons}}}\right),\right.$$

where $\xi_i^{(k)}$ is the $i$th sample of the $k$th constraint model, and $\tau_{\text{cons}}$ is the temperature parameter controlling the approximation to the constraint indicator. While this functionality is in our implementation, our benchmark results use the analytical version.

### A.6 Parallel Expected Hypervolume Improvement

The hypervolume improvement can be computed via the inclusion-exclusion principle, see [13] for details, we focus on the numerical issues concerning qEHVI here. To this end, we define

$$z_{k,i_1,\ldots,i_j}^{(m)} = \min\left[\mathbf{u}_k, \mathbf{f}(\mathbf{x}_{i_1}),\ldots,\mathbf{f}(\mathbf{x}_{i_j})\right],$$

where $\mathbf{f}$ is the vector-valued objective function, and $\mathbf{u}_k$ is the vector of upper bounds of one of $K$ hyper-rectangles that partition the non-Pareto-dominated space, see [13] for details on the partitioning. Letting $\mathbf{l}_k$ be the corresponding lower bounds of the hyper-rectangles, the hypervolume improvement can then be computed as

$$\text{HVI}(\{\mathbf{f}(\mathbf{x}_i)\}_{i=1}^q = \sum_{k=1}^K \sum_{j=1}^q \sum_{X_j \in \mathcal{X}_j} (-1)^{j+1} \prod_{m=1}^M [z_{k,X_j}^{(m)} - l_k^{(m)}]_+, \tag{23}$$

where $\mathcal{X}_j = \{X_j \subset \mathcal{X}_{\text{cand}} : |X_j| = j\}$ is the superset of all subsets of $\mathcal{X}_{\text{cand}}$ of size $j$ and $z_{k,X_j}^{(m)} = z_{k,i_1,\ldots,i_j}^{(m)}$ for $X_j = \{\mathbf{x}_{i_1},\ldots,\mathbf{x}_{i_j}\}$.

To find a numerically stable formulation of the logarithm of this expression, we first re-purpose the $\varphi_{\max}$ function to compute the minimum in the expression of $z_{k,i_1,\ldots,i_j}^{(m)}$, like so $\varphi_{\min}(x) = -\varphi_{\max}(-x)$. Further, we use the $\varphi_+$ function of Sec. A.4 as for the single objective case to approximate $[z_{k,X_j}^{(m)} - l_k^{(m)}]_+$. We then have

$$\log \prod_{m=1}^{M} \varphi_+[z_{k,X_j}^{(m)} - l_k^{(m)}] = \sum_{m=1}^{M} \log \varphi_+[z_{k,X_j}^{(m)} - l_k^{(m)}] \tag{24}$$

Since we can only transform positive quantities to log space, we split the sum in Eq. (23) into positive and negative components, depending on the sign of $(-1)^{j+1}$, and compute the result using a numerically stable implementation of $\log(\exp(\log \text{ of positive terms}) - \exp(\log \text{ of negative terms}))$. The remaining sums over $k$ and $q$ can be carried out by applying `logsumexp` to the resulting quantity. Finally, applying `logsumexp` to reduce over an additional Monte-Carlo sample dimension yields the formulation of qLogEHVI that we use in our multi-objective benchmarks.

### A.7 Probability of Improvement

Numerical improvements for the probability of improvement acquisition which is defined as $\alpha(x) = \Phi\left(\frac{\mu(x)-y^*}{\sigma(x)}\right)$, where $\Phi$ is the standard Normal CDF, can be obtained simply by taking the logarithm using a numerically stable implementation of $\log(\Phi(z)) = \texttt{logerfc}\left(-\frac{1}{\sqrt{2}}z\right) - \log(2)$, where `logerfc` is computed as

$$\texttt{logerfc}(x) = \begin{cases} \log(\texttt{erfc}(x)) & x \leq 0 \\ \log(\texttt{erfcx}(x)) - x^2 & x > 0. \end{cases}$$

### A.8 q-Noisy Expected Improvement

The same numerical improvements used by qLogEI to improve Monte-Carlo expected improvement (qEI) in Appendix A.3 can be applied to improve the fully Monte Carlo Noisy Expected Improvement [6, 52] acquisition function. As in qLogEI, we can (i) approximate the $\max(0, \mathbf{x})$ using a softplus to smooth the sample-level improvements to ensure that they are mathematically positive, (ii) approximate the maximum over the $q$ candidate designs by norm $|| \cdot ||_{\frac{1}{\tau_{\max}}}$, and (iii) take the logarithm to of the resulting smoothed value to mitigate vanishing gradients. To further mitigate vanishing gradients, we can again leverage the Fat Softplus and Fat Maximum approximations. The only notable difference in the $q$EI and $q$NEI acquisition functions is the choice of incumbent, and similarly only a change of incumbent is required to obtain qLogNEI from qLogEI. Specifically, when the scalar $y^*$ in Equation (10) is replaced with a vector containing the new incumbnent under each sample wer obtain the qLogNEI acquisition value. The $i^{\text{th}}$ element of the incumbent vector for qLogNEI is $\max_{j'=q+1}^{n+q} \xi^i(\mathbf{x}_{j'})$, where $\mathbf{x}_{q+1}, \ldots, \mathbf{x}_{n+q}$ are the previously evaluated designs and $\xi^i(\mathbf{x}_{j'})$ is the value of the $j'^{\text{th}}$ point under the $i^{\text{th}}$ sample from the joint posterior over $\mathbf{x}_1, \ldots, \mathbf{x}_{n+q}$. We note that we use a hard maximum to compute the incumbent for each sample because we do not need to compute gradients with respect to the previously evaluated designs $\mathbf{x}_{q+1}, \ldots, \mathbf{x}_{n+q}$.

We note that we obtain computational speed ups by (i) pruning the set of previously evaluated points that considered for being the best incumbent to include only those designs with non-zero probability of being the best design and (ii) caching the Cholesky decomposition of the posterior covariance over the resulting pruned set of previously evaluated designs and using low-rank updates for efficient sampling [14].

For experimental results of $q$LogNEI see Section D.4.

## B Strategies for Optimizing Acquisition Functions

As discussed in Section 2.3, a variety of different approaches and heuristics have been applied to the problem of optimizing acquisition functions. For the purpose of this work, we only consider continuous domains $\mathbb{X}$. While discrete and/or mixed domains are also relevant in practice and have

received substantial attention in recent years – see e.g. Baptista and Poloczek [7], Daulton et al. [15], Deshwal et al. [18], Kim et al. [44], Oh et al. [62], Wan et al. [74] – our work here on improving acquisition functions is largely orthogonal to this (though the largest gains should be expected when using gradient-based optimizers, as is done in mixed-variable BO when conditioning on discrete variables, or when performing discrete or mixed BO using continuous relaxations, probabilistic reparameterization, or straight-through estimators [15]).

Arguably the simplest approach to optimizing acquisition functions is by grid search or random search. While variants of this combined with local descent can make sense in the context of optimizing over discrete or mixed spaces and when acquisition functions can be evaluated efficiently in batch (e.g. on GPUs), this clearly does not scale to higher-dimensional continuous domains due to the exponential growth of space to cover.

Another relatively straightforward approach is to use zeroth-order methods such as `DIRECT` [41] (used e.g. by `Dragonfly` [43]) or the popular CMA-ES [32]. These approaches are easy implement as they avoid the need to compute gradients of acquisition functions. However, not relying on gradients is also what renders their optimization performance inferior to gradient based methods, especially for higher-dimensional problems and/or joint batch optimization in parallel Bayesian optimization.

The most common approach to optimizing acquisition functions on continuous domains is using gradient descent-type algorithms. Gradients are either be computed based on analytically derived closed-form expressions, or via auto-differentiation capabilities of modern ML systems such as PyTorch [63], Tensorflow [1], or JAX [9].

For analytic acquisition functions, a common choice of optimizer is L-BFGS-B [10], a quasi-second order method that uses gradient information to approximate the Hessian and supports box constraints. If other, more general constraints are imposed on the domain, other general purpose nonlinear optimizers such as `SLSQP` [47] or `IPOPT` [73] are used (e.g. by `BoTorch`). For Monte Carlo (MC) acquisition functions, Wilson et al. [77] proposes using stochastic gradient ascent (SGA) based on stochastic gradient estimates obtained via the reparameterization trick [45]. Stochastic first-order algorithms are also used by others, including e.g. Wang et al. [75] and Daulton et al. [15]. Balandat et al. [6] build on the work by Wilson et al. [77] and show how sample average approximation (SAA) can be employed to obtain deterministic gradient estimates for MC acquisition functions, which has the advantage of being able to leverage the improved convergence rates of optimization algorithms designed for deterministic functions such as L-BFGS-B. This general approach has since been used for a variety of other acquisition functions, including e.g. Daulton et al. [13] and Jiang et al. [40].

Very few implementations of Bayesian Optimization actually use higher-order derivative information, as this either requires complex derivations of analytical expressions and their custom implementation, or computation of second-order derivatives via automated differentiation, which is less well supported and computationally much more costly than computing only first-order derivatives. One notable exception is `Cornell-MOE` [78, 79], which supports Newton's method (though this is limited to the acquisition functions implemented in C++ within the library and not easily extensible to other acquisition functions).

### B.1 Common initialization heuristics for multi-start gradient-descent

One of the key issues to deal with gradient-based optimization in the context of optimizing acquisition functions is the optimizer getting stuck in local optima due to the generally highly non-convex objective. This is typically addressed by means of restarting the optimizer from a number of different initial conditions distributed across the domain.

A variety of different heuristics have been proposed for this. The most basic one is to restart from random points uniformly sampled from the domain (for instance, `scikit-optimize` [33] uses this strategy). However, as we have argued in this paper, acquisition functions can be (numerically) zero in large parts of the domain, and so purely random restarts can become ineffective, especially in higher dimensions and with more data points. A common strategy is therefore to either augment or bias the restart point selection to include initial conditions that are closer to "promising points". `GPyOpt` [69] augments random restarts with the best points observed so far, or alternatively points generated via Thompson sampling. `Spearmint` [66] initializes starting points based on Gaussian perturbations of the current best point. `BoTorch` [6] selects initial points by performing Boltzmann sampling on a set of random points according to their acquisition function value; the goal of this strategy is to achieve a

biased random sampling across the domain that is likely to generate more points around regions with high acquisition value, but remains asymptotically space-filling. The initialization strategy used by `Trieste` [64] works similarly to the one in `BoTorch`, but instead of using soft-randomization via Boltzmann sampling, it simply selects the top-$k$ points. Most recently, Gramacy et al. [31] proposed distributing initial conditions using a Delaunay triangulation of previously observed data points. This is an interesting approach that generalizes the idea of initializing "in between" observed points from the single-dimensional case. However, this approach does not scale well with the problem dimension and the number of observed data points due to the complexity of computing the triangulation (with wall time empirically found to be exponential in the dimension, see [31, Fig. 3] and worst-case quadratic in the number of observed points).

However, while these initialization strategies can help substantially with better optimizing acquisition functions, they ultimately cannot resolve foundational issues with acquisition functions themselves. Ensuring that acquisition functions provides enough gradient information (not just mathematically but also numerically) is therefore key to be able to optimize it effectively, especially in higher dimensions and with more observed data points.

## C  Proofs

**Theorem 1.** *Suppose $f$ is drawn from a Gaussian process prior $P_f$, $y^* \leq f^*$, $\mu_n, \sigma_n$ are the mean and standard deviation of the posterior $P_f(f|\mathcal{D}_n)$ and $B \in \mathbb{R}$. Then with probability $1 - \delta$,*

$$P_{\mathbf{x}}\left(\frac{\mu_n(\mathbf{x}) - y^*}{\sigma_n(\mathbf{x})} < B\right) \geq P_{\mathbf{x}}\left(f(\mathbf{x}) < f^* - \epsilon_n\right) \tag{7}$$

*where $\epsilon_n = (f^* - y^*) + \left(\sqrt{-2\log(2\delta)} - B\right)\max_{\mathbf{x}}\sigma_n(\mathbf{x})$.*

*Proof.* We begin by expanding the argument to the $h$ function in Eq. (2) as a sum of 1) the standardized error of the posterior mean $\mu_n$ to the true objective $f$ and 2) the standardized difference of the value of the true objective $f$ at $x$ to the best previously observed value $y^* = \max_i^n y_i$:

$$\frac{\mu_n(\mathbf{x}) - y^*}{\sigma_n(\mathbf{x})} = \frac{\mu_n(\mathbf{x}) - f(\mathbf{x})}{\sigma_n(\mathbf{x})} + \frac{f(\mathbf{x}) - f^*}{\sigma_n(\mathbf{x})} \tag{25}$$

We proceed by bounding the first term on the right hand side. Note that by assumption, $f(\mathbf{x}) \sim \mathcal{N}(\mu_n(\mathbf{x}), \sigma_n(\mathbf{x})^2)$ and thus $(\mu_n(\mathbf{x}) - f(\mathbf{x}))/\sigma_n(\mathbf{x}) \sim \mathcal{N}(0, 1)$. For a positive $C > 0$ then, we use a standard bound on the Gaussian tail probability to attain

$$P\left(\frac{\mu_n(\mathbf{x}) - f(\mathbf{x})}{\sigma_n(\mathbf{x})} > C\right) \leq e^{-C^2/2}/2. \tag{26}$$

Therefore, $(\mu(\mathbf{x}) - f(\mathbf{x}))/\sigma_n(\mathbf{x}) < C$ with probability $1 - \delta$ if $C = \sqrt{-2\log(2\delta)}$.

Using the bound just derived, and forcing the resulting upper bound to be less than $B$ yields a sufficient condition to imply $\mu_n(\mathbf{x}) - y^* < B\sigma_n(\mathbf{x})$:

$$\frac{\mu_n(\mathbf{x}) - y^*}{\sigma_n(\mathbf{x})} \leq C + \frac{f(\mathbf{x}) - y^*}{\sigma_n(\mathbf{x})} < B \tag{27}$$

Letting $f^* := f(\mathbf{x}^*)$, re-arranging and using $y^* = f^* + (y^* - f^*)$ we get with probability $1 - \delta$,

$$f(\mathbf{x}) \leq f^* - (f^* - y^*) - (\sqrt{-2\log(2\delta)} - B)\sigma_n(\mathbf{x}). \tag{28}$$

Thus, we get

$$P_{\mathbf{x}}\left(\frac{\mu_n(\mathbf{x}) - y^*}{\sigma_n(\mathbf{x})} < B\right) \geq P_{\mathbf{x}}\left(f(\mathbf{x}) \leq f^* - (f^* - y^*) - (\sqrt{-2\log(2\delta)} - B)\sigma_n(x)\right)$$

$$\geq P_x\left(f(\mathbf{x}) \leq f^* - (f^* - y^*) - (\sqrt{-2\log(2\delta)} - B)\max_{\mathbf{x}}\sigma_n(\mathbf{x})\right). \tag{29}$$

Note that the last inequality gives a bound that is not directly dependent on the evaluation of the posterior statistics of the surrogate at any specific $\mathbf{x}$. Rather, it is dependent on the optimality gap $f^* - y^*$ and the maximal posterior standard deviation, or a bound thereof. Letting $\epsilon_n = (f^* - y^*) - (\sqrt{-2\log(2\delta)} - B)\max_{\mathbf{x}}\sigma_n(\mathbf{x})$ finishes the proof. $\qquad\square$

**Lemma 2.** *[Relative Approximation Guarantee] Given $\tau_0, \tau_{\max} > 0$, the approximation error of* qLogEI *to* qEI *is bounded by*

$$\left|e^{\text{qLogEI}(\mathbf{X})} - \text{qEI}(\mathbf{X})\right| \le (q^{\tau_{\max}} - 1)\,\text{qEI}(\mathbf{X}) + \log(2)\tau_0 q^{\tau_{\max}}. \tag{11}$$

*Proof.* Let $z_{iq} = \xi_i(\mathbf{x}_q) - y^*$, where $i \in \{1, ..., n\}$, and for brevity of notation, and let $\texttt{lse}$, $\texttt{lsp}$ refer to the $\texttt{logsumexp}$ and $\texttt{logsoftplus}$ functions, respectively, and $\text{ReLU}(x) = \max(x, 0)$. We then bound $n|e^{\text{qLogEI}(\mathbf{X})} - \text{qEI}(\mathbf{X})|$ by

$$
\begin{aligned}
&\left|\exp(\texttt{lse}_i(\tau_{\max}\texttt{lse}_q(\texttt{lsp}_{\tau_0}(z_{iq})/\tau_{\max}))) - \sum_i \max_q \text{ReLU}(z_{iq})\right| \\
&\le \sum_i \left|\exp(\tau_{\max}\texttt{lse}_q(\texttt{lsp}_{\tau_0}(z_{iq})/\tau_{\max})) - \max_q \text{ReLU}(z_{iq})\right| \\
&= \sum_i \left|\|\texttt{softplus}_{\tau_0}(z_{i\cdot})\|_{1/\tau_{\max}} - \max_q \text{ReLU}(z_{iq})\right| \\
&\le \sum_i \left|\|\texttt{softplus}_{\tau_0}(z_{i\cdot})\|_{1/\tau_{\max}} - \max_q \texttt{softplus}_{\tau_0}(z_{iq})\right| \\
&\quad + \left|\max_q \texttt{softplus}_{\tau_0}(z_{iq}) - \max_q \text{ReLU}(z_{iq})\right|
\end{aligned}
\tag{30}
$$

First and second inequalities are due to the triangle inequality, where for the second we used $|a - c| \le |a - b| + |b - c|$ with $b = \max_q \texttt{softplus}(z_{iq})$.

To bound the first term in the sum, note that $\|\mathbf{x}\|_\infty \le \|\mathbf{x}\|_q \le \|\mathbf{x}\|_\infty d^{1/q}$, thus $0 \le (\|\mathbf{x}\|_q - \|\mathbf{x}\|_\infty) \le d^{1/q} - 1\|\mathbf{x}\|_\infty$, and therefore

$$
\begin{aligned}
\left|\|\texttt{softplus}_{\tau_0}(z_{i\cdot})\|_{1/\tau_{\max}} - \max_q \texttt{softplus}_{\tau_0}(z_{iq})\right| &\le (q^{\tau_{\max}} - 1)\max_q \texttt{softplus}_{\tau_0}(z_{iq}) \\
&\le (q^{\tau_{\max}} - 1)(\max_q \text{ReLU}(z_{iq}) + \log(2)\tau_0)
\end{aligned}
$$

The second term in the sum can be bound due to $|\texttt{softplus}_{\tau_0}(x) - \text{ReLU}(x)| \le \log(2)\tau_0$ (see Lemma 7 below) and therefore,

$$\left|\max_q \texttt{softplus}_{\tau_0}(z_{iq}) - \max_q \text{ReLU}_{\tau_0}(z_{iq})\right| \le \log(2)\tau_0.$$

Dividing Eq. (30) by $n$ to compute the sample mean finishes the proof for the Monte-Carlo approximations to the acquisition value. Taking $n \to \infty$ further proves the result for the mathematical definitions of the parallel acquisition values, i.e. Eq. (4). $\qquad\square$

Approximating the ReLU using the $\texttt{softplus}_\tau(x) = \tau \log(1 + \exp(x/\tau))$ function leads to an approximation error that is at most $\tau$ in the infinity norm, i.e. $\|\texttt{softplus}_\tau - \text{ReLU}\|_\infty = \log(2)\tau$. The following lemma formally proves this.

**Lemma 7.** *Given $\tau > 0$, we have for all $x \in \mathbb{R}$,*

$$|softplus_\tau(x) - ReLU(x)| \le \log(2)\,\tau. \tag{31}$$

*Proof.* Taking the (sub-)derivative of $\texttt{softplus}_\tau - \text{ReLU}$, we get

$$\partial_x \texttt{softplus}_\tau(x) - \text{ReLU}(x) = (1 + e^{-x/\tau})^{-1} - \begin{cases} 1 & x > 0 \\ 0 & x \le 0 \end{cases}$$

which is positive for all $x < 0$ and negative for all $x > 0$, hence the extremum must be at $x$, at which point $\texttt{softplus}_\tau(0) - \text{ReLU}(0) = \log(2)\tau$. Analyzing the asymptotic behavior, $\lim_{x \to \pm\infty}(\texttt{softplus}_\tau(x) - \text{ReLU}(x)) = 0$, and therefore $\texttt{softplus}_\tau(x) > \text{ReLU}(x)$ for $x \in \mathbb{R}$. $\quad\square$

Approximation guarantees for the fat-tailed non-linearities of App. A.4 can be derived similarly.

# D  Additional Empirical Details and Results

## D.1  Experimental details

All algorithms are implemented in BoTorch. The analytic EI, qEI, cEI utilize the standard BoTorch implementations. We utilize the original authors' implementations of single objective JES [36], GIBBON [61], and multi-objective JES [71], which are all available in the main BoTorch repository. All simulations are ran with 32 replicates and error bars represent $\pm 2$ times the standard error of the mean. We use a Matern-5/2 kernel with automatic relevance determination (ARD), i.e. separate length-scales for each input dimension, and a top-hat prior on the length-scales in $[0.01, 100]$. The input spaces are normalized to the unit hyper-cube and the objective values are standardized during each optimization iteration.

## D.2  Additional Empirical Results on Vanishing Values and Gradients

The left plot of Figure 1 in the main text shows that for a large fraction of points across the domain the gradients of EI are numerically essentially zero. In this section we provide additional detail on these simulations as well as intuition for results.

The data generating process (DGP) for the training data used for the left plot of Figure 1 is the following: 80% of training points are sampled uniformly at random from the domain, while 20% are sampled according to a multivariate Gaussian centered at the function maximum with a standard deviation of 25% of the length of the domain. The idea behind this DGP is to mimic the kind of data one would see during a Bayesian Optimization loop (without having to run thousands of BO loops to generate the Figure 1). Under this DGP with the chosen test problem, the incumbent (best observed point) is typically better than the values at the random test locations, and this becomes increasingly the case as the dimensionality of the problem increases and the number of training points grows. This is exactly the situation that is typical when conducting Bayesian Optimization.

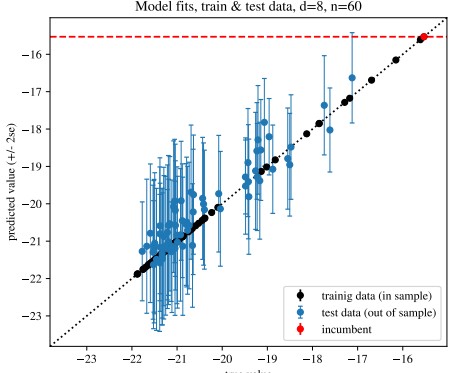

Figure 15: Model fits for a typical replicate used in generating the Fig. 1 (left). While there is ample uncertainty in the test point predictions (blue, chosen uniformly at random), the mean prediction for the majority of points is many standard deviations away from the incumbent value (red).

Figure 16: Histogram of $z(x)$, the argument to $h$ in (2), corresponding to Figure 15. Vertical lines are the thresholds corresponding to values $z$ below which $h(z)$ is less than the respective threshold. The majority of the test points fall below these threshold values.

For a particular replicate, Figure 15 shows the model fits in-sample (black), out-of-sample (blue), and the best point identified so far (red) with 60 training points and a random subset of 50 (out of 2000) test points. One can see that the model produces decent mean predictions for out-of-sample data, and that the uncertainty estimates appear reasonably well-calibrated (e.g., the credible intervals typically cover the true value). A practitioner would consider this a good model for the purposes of Bayesian Optimization. However, while there is ample uncertainty in the predictions of the model away from the training points, for the vast majority of points, the mean prediction is many standard deviations away from the incumbent value (the error bars are $\pm$ 2 standard deviations). This is the key reason for EI taking on zero (or vanishingly small) values and having vanishing gradients.

To illustrate this, Figure 16 shows the histogram of $z(x)$ values, the argument to the function $h$ in (2). It also contains the thresholds corresponding to the values $z$ below which $h(z)$ is less than the respective threshold. Since $\sigma(x)$ is close to 1 for most test points (mean: 0.87, std: 0.07), this is more or less the same as saying that $\mathrm{EI}(z(x))$ is less than the threshold. It is evident from the histogram that the majority of the test points fall below these threshold values (especially for larger thresholds), showing that the associated acquisition function values (and similarly the gradients) are numerically almost zero and causing issues during acquisition function optimization.

## D.3   Parallel Expected Improvement

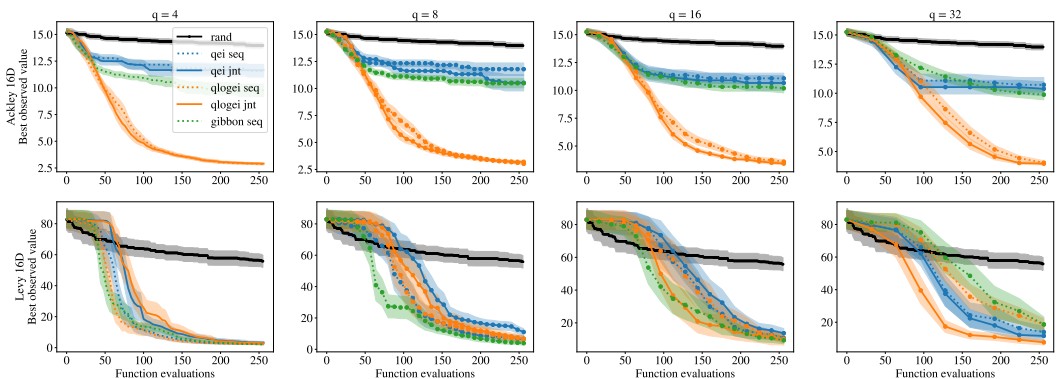

Figure 17: Parallel optimization performance on the Ackley and Levy functions in 16 dimensions. qLogEI outperforms all baselines on Ackley, where joint optimization of the batch also improves on the sequential greedy. On Levy, joint optimization of the batch with qLogEI starts out performing worse in terms of BO performance than sequential, but wins out over all sequential methods as the batch size increases.

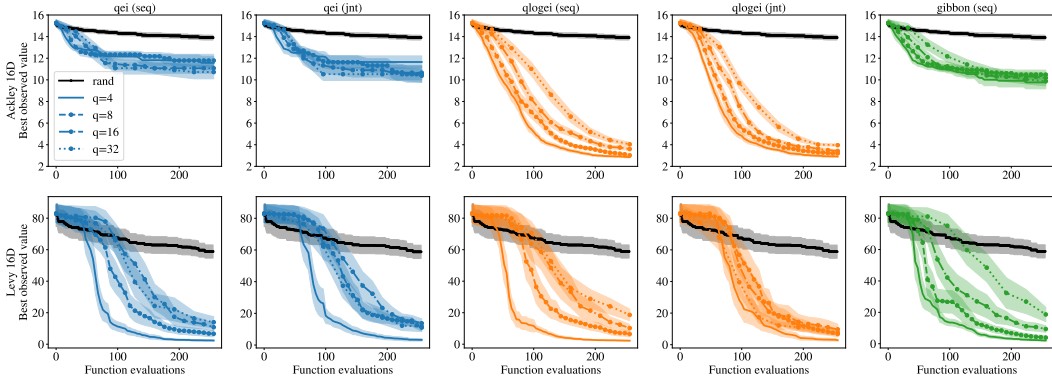

Figure 18: Breakdown of the parallel optimization performance of Figure 17 per method, rather than per batch size. On Levy, qLogEI exhibits a much smaller deterioration in BO performance due to increases in parallelism than the methods relying on sequentially optimized batches of candidates.

Figure 17 reports optimization performance of parallel BO on the 16-dimensional Ackley and Levy functions for both sequential greedy and joint batch optimization. Besides the apparent substantial advantages of qLogEI over qEI on Ackley, a key observation here is that jointly optimizing the candidates of batch acquisition functions can yield highly competitive optimization performance, especially as the batch size increases. Notably, joint optimization of the batch with qLogEI starts out performing worse in terms of BO performance than sequential on the Levy function, but outperforms all sequential methods as the batch size increases. See also Figure 18 for the scaling of each method with respect to the batch size $q$.

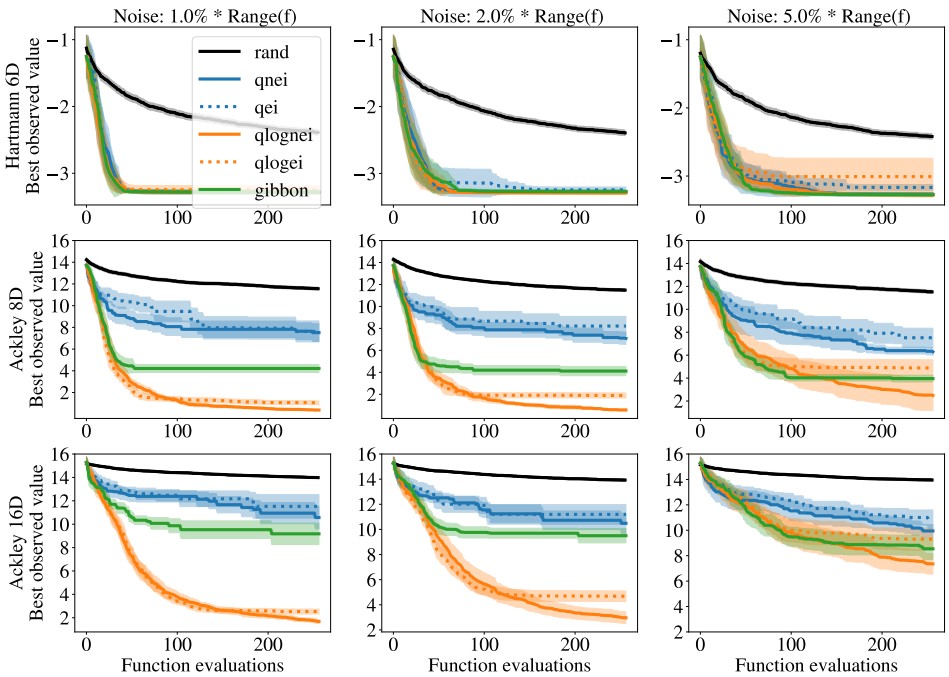

Figure 19: Optimization performance with noisy observations on Hartmann 6d (top), Ackley 8d (mid), and Ackley 16 (bottom) for varying noise levels and $q = 1$. We set the noise level as a proportion of the maximum range of the function, which is $\approx 3.3$ for Hartmann and $\approx 20$ for Ackley. That is, the $1\%$ noise level corresponds to a standard deviation of $0.2$ for Ackley. qLogNEI outperforms both canonical EI counterparts and Gibbon significantly in most cases, especially in higher dimensions.

### D.4 Noisy Expected Improvement

Figure 19 benchmarks the "noisy" variant, qLogNEI. Similar to the noiseless case, the advantage of the LogEI versions over the canonical counterparts grows as with the dimensionality of the problem, and the noisy version improves on the canonical versions for larger noise levels.

### D.5 Multi-Objective optimization with qLogEHVI

Figure 20 compares qLogEHVI and qEHVI on 6 different test problems with 2 or 3 objectives, and ranging from 2-30 dimensions. This includes 3 real world inspired problems: cell network design for optimizing coverage and capacity [19], laser plasma acceleration optimization [38], and vehicle design optimization [54, 68]. The results are consistent with our findings in the single-objective and constrained cases: qLogEHVI consistently outperforms qEHVI, and the gap is larger on higher dimensional problems.

### D.6 Combining LogEI with `TuRBO` for High-Dimensional Bayesian Optimization

In the main text, we show how LogEI performs particularly well relative to other baselines in high dimensional spaces. Here, we show how LogEI can work synergistically with trust region based methods for high-dimensional BO, such as TuRBO [24].

Fig. 21 compares the performance of LogEI, TuRBO-1 + LogEI, TuRBO-1 + EI, as well as the original Thompson-sampling based implementation for the 50d Ackley test problem. Combining TuRBO-1 with LogEI results in substantially better performance than the baselines when using a small number of function evaluations. Thompson sampling (TS) ultimately performs better after $10,000$ evaluations, but this experiment shows the promise of combining TuRBO and LogEI in settings where we cannot do thousands of function evaluations. Since we optimize batches of $q = 50$ candidates jointly, we also increase the number of Monte-Carlo samples from the Gaussian process

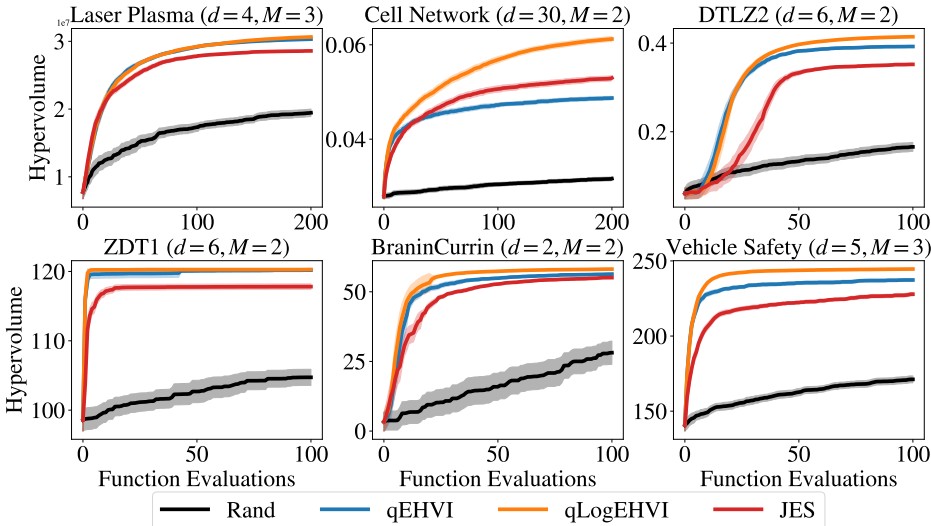

Figure 20: Sequential ($q = 1$) optimization performance on multi-objective problems, as measured by the hypervolume of the Pareto frontier across observed points. This plot includes JES [71]. Similar to the single-objective case, qLogEHVI significantly outperforms all baselines on all test problems.

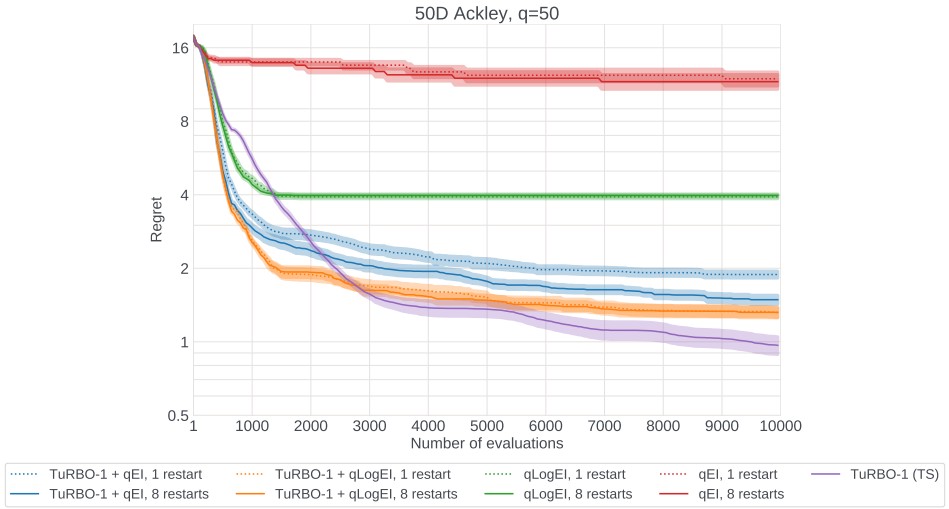

Figure 21: Combining LogEI with TuRBO on the high-dimensional on the 50d Ackley problem yields significant improvement in objective value for a small number of function evaluations. Unlike for qEI, no random restarts are necessary to achieve good performance when performing joint optimization of the batch with qLogEI ($q = 50$).

from $128$, the BoTorch default, to $512$, and use the fat-tailed smooth approximations of Sec. A.4 to ensure a strong gradient signal to all candidates of the batch.

Regarding the ultimate out-performance of TS over qLogEI, we think this is due to a model mis-specification since the smoothness of a GP with the Matern-5/2 kernel cannot express the non-differentiability of Ackley at the optimum.

## D.7 Constrained Problems

While running the benchmarks using CEI in section 5, we found that we in fact improved upon a best known result from the literature. We compare with the results in Coello and Montes [11], which are generated using 30 runs of **80,000 function evaluations** each.

- For the pressure vessel design problem, Coello and Montes [11] quote a best-case feasible objective of 6059.946341. Out of just 16 different runs, LogEI achieves a worst-case feasible objective of 5659.1108 **after only 110 evaluations**, and a best case of 5651.8862, a notable reduction in objective value using almost three orders of magnitude fewer function evaluations.

- For the welded beam problem, Coello and Montes [11] quote 1.728226, whereas LogEI found a best case of 1.7496 after 110 evaluations, which is lightly worse, but we stress that this is using three orders of magnitude fewer evaluations.

- For the tension-compression problem, LogEI found a feasible solution with value 0.0129 after 110 evaluations compared to the 0.012681 reported in in [11].

We emphasize that genetic algorithms and BO are generally concerned with distinct problem classes: BO focuses heavily on sample efficiency and the small-data regime, while genetic algorithms often utilize a substantially larger number of function evaluations. The results here show that in this case BO is competitive with and can even outperform a genetic algorithm, using only a tiny fraction of the sample budget, see App. D.7 for details. Sample efficiency is particularly relevant for physical simulators whose evaluation takes significant computational effort, often rendering several tens of thousands of evaluations infeasible.

### D.8 Parallel Bayesian Optimization with cross-batch constraints

In some parallel Bayesian optimization settings, batch optimization is subject to non-trivial constraints across the batch elements. A natural example for this are budget constraints. For instance, in the context of experimental material science, consider the case where each manufactured compound requires a certain amount of different materials (as described by its parameters), but there is only a fixed total amount of material available (e.g., because the stock is limited due to cost and/or storage capacity). In such a situation, batch generation will be subject to a budget constraint that is not separable across the elements of the batch. Importantly, in that case sequential greedy batch generation is not an option since it is not able to incorporate the budget constraint. Therefore, joint batch optimization is required.

Here we give one such example in the context of Bayesian Optimization for sequential experimental design. We consider the five-dimensional silver nanoparticle flow synthesis problem from Liang et al. [53]. In this problem, to goal is to optimize the absorbance spectrum score of the synthesized nanoparticles over five parameters: four flow rate ratios of different components (silver, silver nitrate, trisodium citrate, polyvinyl alcohol) and a total flow rate $Q_{tot}$.

The original problem was optimized over a discrete set of parameterizations. For our purposes we created a continuous surrogate model based on the experimental dataset (available from `https://github.com/PV-Lab/Benchmarking`) by fitting an RBF interpolator (smoothing factor of 0.01) in `scipy` on the (negative) loss. We use the same search space as Liang et al. [53], but in addition to the box bounds on the parameters we also impose an additional constraint on the total flow rate $Q_{tot}^{max} = 2000$ μL/min across the batch: $\sum_{i=1}^{q} Q_{tot}^i \leq Q_{tot}^{max}$ (the maximum flow rate per syringe / batch element is 1000μL/min). This constraint expresses the maximum throughput limit of the microfluidic experimentation setup. The result of this constraint is that we cannot consider the batch elements (in this case automated syringe pumps) have all elements of a batch of experiments operate in the high-flow regime at the same time.

In our experiment, we use a batch size of $q = 3$ and start the optimization from 5 randomly sampled points from the domain. We run 75 replicates with random initial conditions (shared across the different methods), error bars show $\pm$ two times the standard error of the mean. Our baseline is uniform random sampling from the domain (we use a hit-and-run sampler to sample uniformly from the constraint polytope $\sum_{i=1}^{q} Q_{tot}^i \leq Q_{tot}^{max}$). We compare qEI vs. qLogEI, and for each of the two we evaluate (i) the version with the batch constraint imposed explicitly in the optimizer (the optimization in this case uses `scipy`'s SLSQP solver), and (ii) a heuristic that first samples the total flow rates $\{Q_{tot}^i\}_{i=1}^{q}$ uniformly from the constraint set, and then optimizes the acquisition function with the flow rates fixed to the sampled values.

The results in Figure 22 show that while both the heuristic ("random $Q_{tot}$") and the proper constrained optimization ("batch-constrained $Q_{tot}$") substantially outperform the purely random baseline, it requires uisng both LogEI *and* proper constraints to achieve additional performance gains over the other 3 combinations. Importantly, this approach is only possible by performing joint optimization of

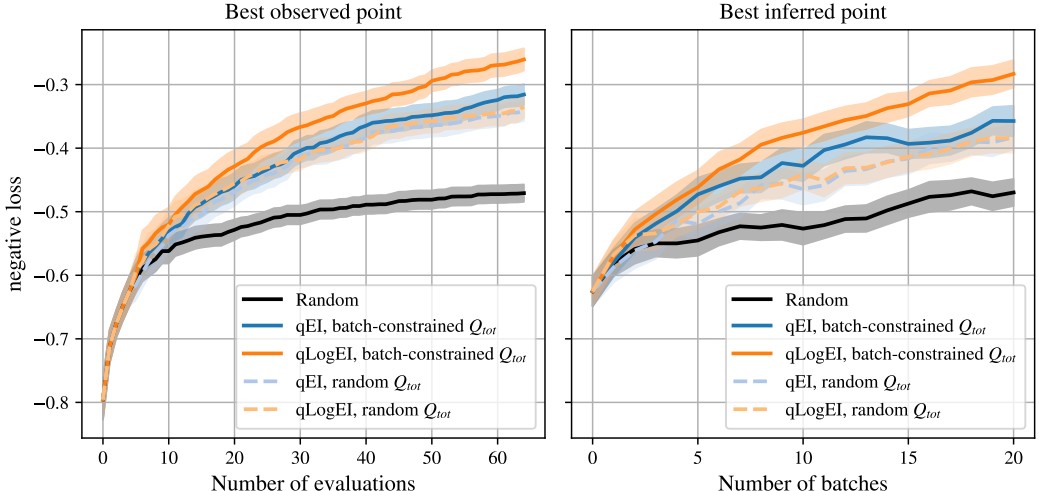

Figure 22: Optimization results on the nanomaterial synthesis material science problem with cross-batch constraints. While qLogEI outperforms qEI under the proper constrained ("batch-constrained $Q_{tot}$") optimization, this is not the case for the the heuristic ("random $Q_{tot}$"), demonstrating the value of both joint batch optimization with constraints and LogEI.

the batch, which underlines the importance of qLogEI and its siblings being able to achieve superior joint batch optimization in settings like this.

### D.9  Details on Multi-Objective Problems

We consider a variety of multi-objective benchmark problems. We evaluate performance on three synthetic biobjective problems Branin-Currin ($d = 2$) [8], ZDT1 ($d = 6$) [83], and DTLZ2 ($d = 6$) [17]. As described in 5, we also evaluated performance on three real world inspired problems. For the laser plasma acceleration problem, we used the public data available at Irshad et al. [39] to fit an independent GP surrogate model to each objective. We only queried te surrogate at the highest fidelity to create a single fidelity benchmark.

### D.10  Effect of Temperature Parameter

In Figure 23, we examine the effect of fixed $\tau$ for the softplus operator on optimization performance. We find that smaller values typically work better.

### D.11  Effect of the initialization strategy

Packages and frameworks commonly utilize smart initialization heuristics to improve acquisition function optimization performance. In Figure 24, we compare simple random restart optimization, where initial points are selected uniformly at random, with BoTorch's default initialization strategy, which evaluates the acquisition function on a large number of points selected from a scrambled Sobol sequence, and selects $n$ points at random via Boltzman sampling (e.g., sampling using probabilities computed by taking a softmax over the acquisition values [6]. Here we consider 1024 initial candidates. We find that the BoTorch initialization strategy improves regret for all cases, and that qLogEI, followed by UCB show less sensitivity to the choice of initializations strategy. Figure25 examines the sensitivity of qEI to the number of initial starting points when performing standard random restart optimization and jointly optimizing the $q$ points in the batch. We find that, consistent with our empirical and theoretical results in the main text, qEI often gets stuck in local minima for the Ackley test function, and additional random restarts often improve results but do not compensate for the fundamental optimality gap. The performance of qLogEI also improves as the number of starting points increases.

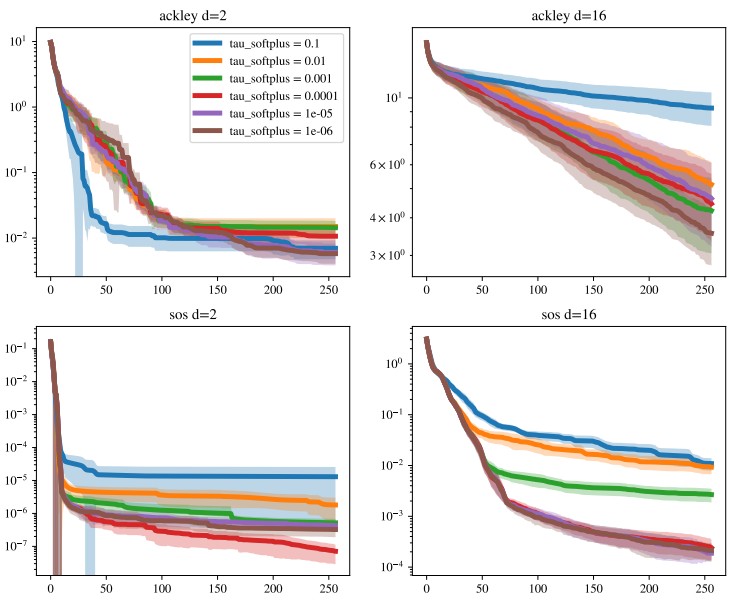

Figure 23: Ablation study on the convergence characteristics of LogEI on Ackley and sum of squares (SOS) problems in 2 and 16 dimensions. The study shows that it is important to choose a small $\tau_0$ for the best convergence properties, which results in a very tight approximation to the original `ReLU` non-linearity in the integrand. Critically, setting $\tau_0$ as low as $10^{-6}$ is only possible due to the transformation of all computations into log-space. Otherwise, the smoothed acquisition utility would exhibit similarly numerically vanishing gradients as the original `ReLU` non-linearity.

| | CELL NETWORK | BRANIN-CURRIN | DTLZ2 | LASER PLASMA | ZDT1 | VEHICLE SAFETY |
|---|---|---|---|---|---|---|
| JES | 21.6 (+/- 1.1) | 89.6 (+/- 3.3) | 33.6 (+/- 1.0) | 57.3 (+/- 0.7) | 72.7 (+/- 1.0) | 47.0 (+/- 1.6) |
| QEHVI | 0.6 (+/- 0.0) | 0.7 (+/- 0.0) | 1.0 (+/- 0.0) | 3.0 (+/- 0.1) | 0.6 (+/- 0.0) | 0.6 (+/- 0.0) |
| QLOGEHVI | 9.2 (+/- 0.8) | 10.0 (+/- 0.4) | 5.8 (+/- 0.2) | 31.6 (+/- 1.7) | 7.2 (+/- 0.7) | 2.1 (+/- 0.1) |
| RAND | 0.2 (+/- 0.0) | 0.2 (+/- 0.0) | 0.2 (+/- 0.0) | 0.3 (+/- 0.0) | 0.3 (+/- 0.0) | 0.3 (+/- 0.0) |

Table 1: Multi-objectve acquisition function optimization wall time in seconds on CPU (2x Intel Xeon E5-2680 v4 @ 2.40GHz). We report the mean and $\pm$ 2 standard errors.

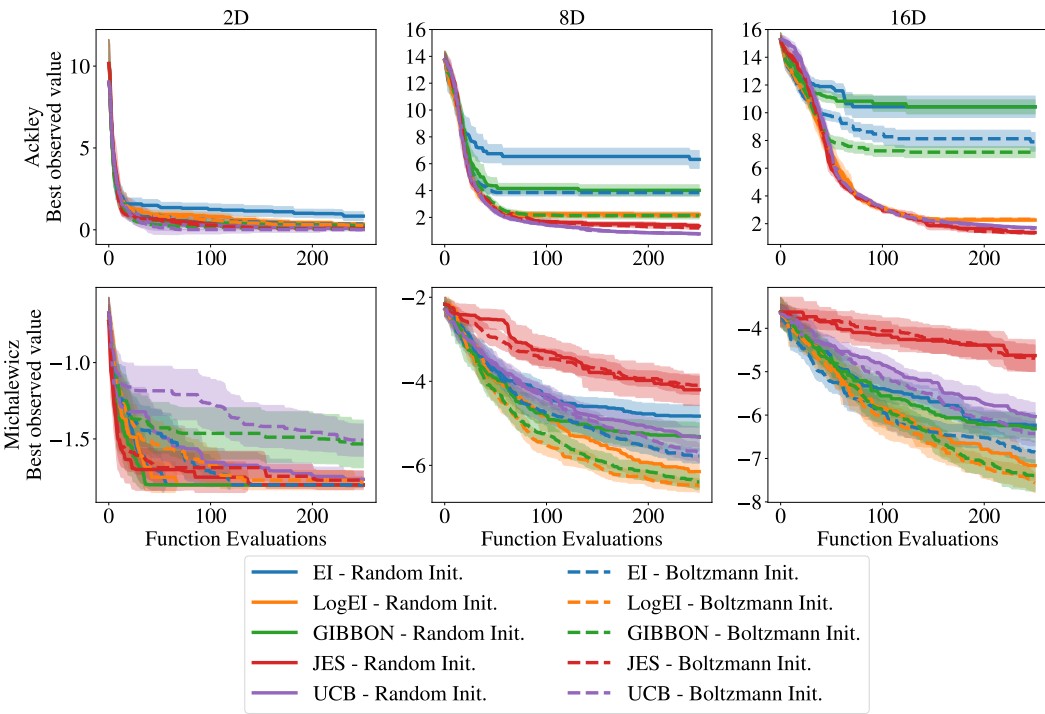

Figure 24: Sensitivity to the initialization strategy. Random selects random restart points from the design space uniformly at random, whereas Boltzmann initialization is the default BoTorch initialization strategy which selects points with higher acquisition function values with a higher probability via Boltzmann sampling.

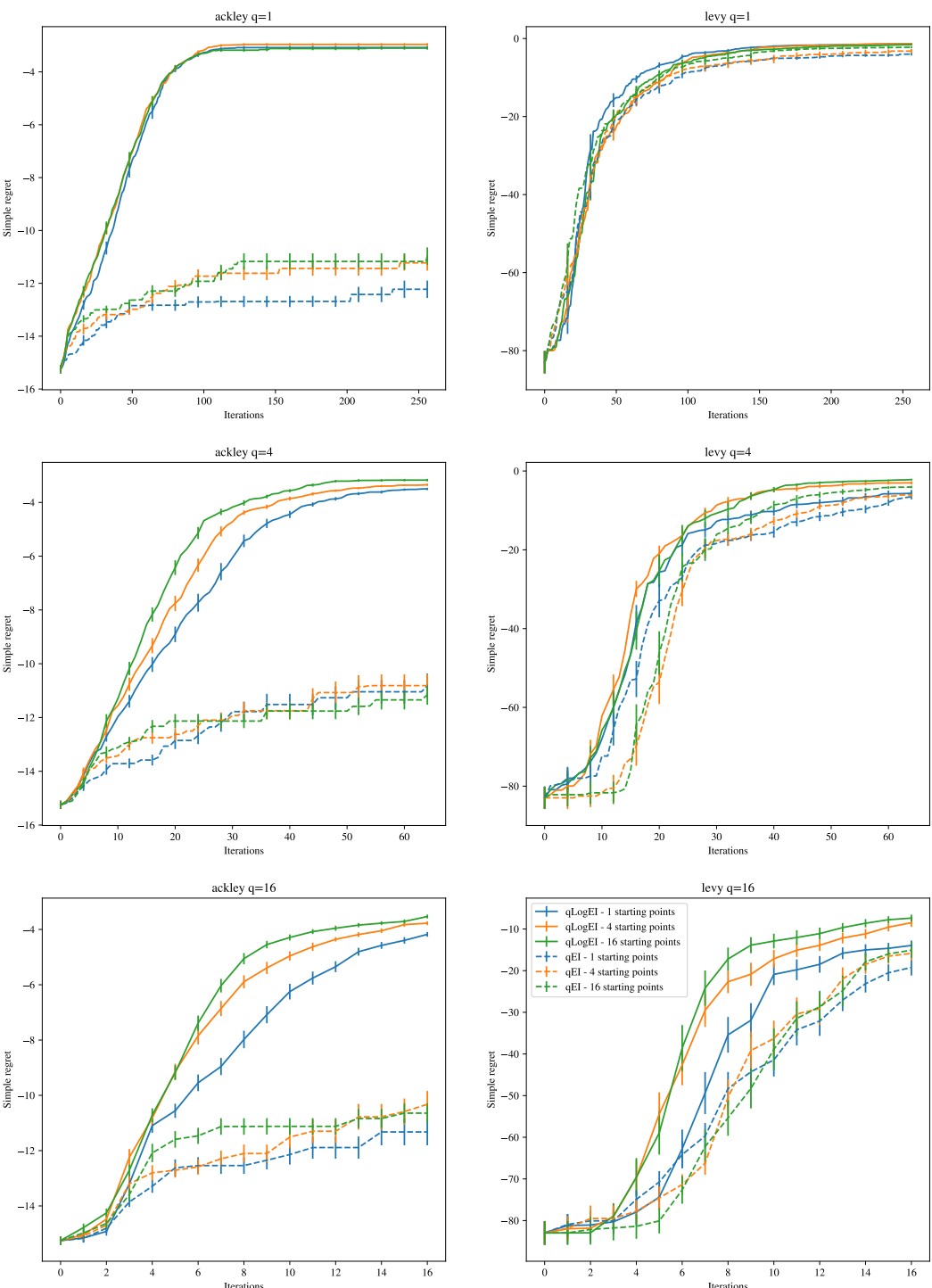

Figure 25: Sensitivity to number of starting points with multi-start optimization for the 16D Ackley and Levy test problems. Note: We plot negative regret, so higher is better.

