# OpenReview forum: "Unexpected Improvements to Expected Improvement for Bayesian Optimization"
_NeurIPS.cc/2023/Conference — NeurIPS 2023 spotlight_

### Official Review · Reviewer_UCkN · 2023-06-25

**Soundness:** 4 excellent
**Presentation:** 4 excellent
**Contribution:** 4 excellent
**Rating:** 8
**Confidence:** 5

**Summary:**

This paper addresses a major weakness of Bayesian expected improvement acquisition functions, which are ubiquitously used for black-box optimization tasks such as computational hyperparameter tuning, materials science, and biomedical research. It is very common to use gradient-based optimizers to find local maxima of the acquisition surface, however expected-improvement acquisition functions have the very unfortunate pathology of a completely flat acquisition surface (where both the acquisition value and acquisition gradient are 0) on large regions of input space, particularly as optimization progresses and the best known solution improves. This pathology makes Bayesian optimization extremely sensitive to implementation decisions, particularly the initialization scheme of the acquisition maximization subproblem, which hinders BayesOpt practitioners in academia and industry. This paper rightfully places numerical precision and stability as one of the primary considerations in acquisition function design, and proposes simple, intuitive modifications to expected improvement acquisition functions that significantly improve performance.

**Strengths:**

I am strongly in favor of accepting this paper. The basic problem the authors are addressing is one I have often encountered myself, and I have even tried some similar ideas as those presented in this paper to try to address the problem, however I had to shelve the project due to competing demands for my time. I'm delighted to see the problem addressed so thoroughly here.

The greatest strength of this paper is the emphasis placed on how acquisition function design interacts with the optimization algorithms used to find their maxima. Generally speaking I feel this aspect of acquisition function design is often neglected in many Bayesian optimization papers, to the great detriment of the field.

Given the widespread use of BayesOpt across industries, and the use of EI-style acquisition functions in particular, I think this paper could have significant practical impact on multiple fields.

**Weaknesses:**

This work is ready for publication without any significant revisions. I would encourage reviewers in general to think about the opportunity cost of burdening authors with minor or tangential concerns, slowing the development of follow-up work.

It's worth noting that the weaknesses of EI-style acquisition functions are fairly well documented in latent-space BayesOpt papers, such as [1] and [2]. Both of those works employed a heuristic I didn't see mentioned in the paper, which is to scale the max_{x_i \in D} f(x_i) term in the acquisition function with some factor < 1 (e.g. 0.9). Some brief discussion in the related work on this point could help better communicate the potential impact of this paper.

[1] Tripp, A., Daxberger, E., and Hernandez-Lobato, J. M. ´
Sample-efficient optimization in the latent space of deep
generative models via weighted retraining. Advances in
Neural Information Processing Systems, 33, 2020.

[2] Stanton, S., Maddox, W., Gruver, N., Maffettone, P., Delaney, E., Greenside, P., & Wilson, A. G. (2022, June). Accelerating bayesian optimization for biological sequence design with denoising autoencoders. In International Conference on Machine Learning (pp. 20459-20478). PMLR.

**Questions:**

Can you comment on whether you expect batch acquisition value optimization to outperform sequential greedy optimization when there are strong locality constraints placed on the inner loop problem (i.e. d(x_0, x_t) < \varepsilon for all x_t optimization iterates)? Intuitively it seems that the performance of batch acquisition optimization once again comes down to heuristics for choosing the right collection of points as the initial solution (e.g. [2]), since the iterates may not be able to move far enough from the initialization for the improved acquisition landscape to make much difference.

**Limitations:**

I think the discussion section could be expanded a bit. In particular I think the following rather vague sentence could be made more specific:

"While our contributions may not apply verbatim to other classes of acquisition functions, our key insights and strategies do translate and could help e.g. with improving information-based [20, 42], cost-aware [26, 36], and other types of acquisition functions that are prone to similar numerical challenges."

I take this to mean that this paper has primarily focused on resolving numerical difficulties arising from the use of the max operator, and other acquisition functions may have numerical issues from operators that are not the max. If this is the case I think it could be stated more clearly, as it would give readers a clearer picture of avenues for future work.

---

> ### Author Rebuttal · Authors · 2023-08-10
>
> Thank you for your detailed review and encouragement.
>
> __Regarding the scaling of the incumbent__
>
> Thank you for pointing us to the heuristic of scaling the incumbent by a factor. We have encountered and experimented with this heuristic in the past, and while it can be employed to try to avoid numerical degeneracies, it has drawbacks compared to the solution proposed in the present paper:
> - For a fixed scaling factor, the resulting acquisition function – to the best of our knowledge –  does not have a principled grounding, requires setting a hyper-parameter that the optimization is sensitive to, and it is unclear which scaling factor is sufficient to remedy the numerical problems a-priori.
> - Using a homotopy approach to sequentially increase the scaling factor can be more robust, but this requires solving a sequence of acquisition function optimization problems, which is more computationally expensive than optimizing LogEI once. We experimented with this approach in the early stages of this project.
>
> LogEI successfully circumvents the downsides of the incumbent-scaling approach without adding the computational overhead of solving multiple acquisition function optimization problems.
>
> We agree that discussing this will further improve the paper and the motivation of the methods. Will do so in the related work section.
>
> __Regarding greedy vs sequential batch optimization__
>
> We are not sure we fully understand your comment regarding “strong locality constraints”, but believe it may refer to explicitly constraining the distance between new candidates and previously seen points in the context of latent space BO. Such a heuristic has been used in the literature to avoid exploring parts of the latent space that the decoder cannot map back well in the original input space. Assuming this is indeed the setting, it isn’t immediately clear to us whether these imposed locality constraints would have a significant effect on whether joint optimization is beneficial over sequential greedy. Certainly, as $\varepsilon \to 0$ we would expect the benefit to disappear (and similarly we would expect to recover it fully as $\varepsilon -> \infty$, assuming the decoder performance is good across the entire latent space), but it’s hard to say in general what the behavior would be. It’s an interesting question that deserves further study.
>
> __Regarding the extensibility of the methods__
>
> We seek to highlight two distinct aspects:
>
> - Analytical LogEI relies primarily on careful and stable implementations of Gaussian log-probabiliby functions, their sums, differences, and fractions. We believe information-based acquisition functions like Gibbon and Joint Entropy Search (JES) could benefit from similar treatments. A particularly striking result of our experiments revealed that JES fails to outperform random search on the Michaelwicz function in greater than 8 dimensions, which could be caused by numerical issues of the acquisition function.
>
> - Monte-Carlo (“q”) LogEI primarily relies on novel smooth approximations to the ReLU non-linearity, as well as max operator, in addition to the associated log-tricks. We believe that especially the fat-tailed smooth operators (Appendix A.3) could find wider application even outside of Bayesian optimization, in the design and optimization of deep architectures, which are known to lead to saturating nonlinearities and vanishing gradients. We aim to explore this in follow up work.

---

> > ### Comment · Reviewer_UCkN · 2023-08-11
> > **Acknowledgement**
> >
> > Thanks for your response, I remain strongly in favor of acceptance. I feel that it is relevant to mention that the problem the authors solve has been such a headache for me that I've already started using the code the authors included in the supplementary material for my own work. I can't think of better evidence for the potential impact of the paper.
> >
> > I completely agree with your comments on the incumbent scaling approach. I will have to think a bit more about how to make my question regarding constrained batch optimization more clear. Your clarification of the discussion section is great, I hope you include it in the camera-ready.

---

> > > ### Author Response · Authors · 2023-08-12
> > >
> > > We will include the additional clarification in the paper, and are happy to hear you are already using the code!

---

### Official Review · Reviewer_u8yk · 2023-07-04

**Soundness:** 4 excellent
**Presentation:** 4 excellent
**Contribution:** 3 good
**Rating:** 7
**Confidence:** 5

**Summary:**

The paper proposes LogEI, family of acquisition functions with improved numerical stability over EI that makes it more suitable for gradient-based acquisition function optimization, all while retaining similar optima as EI. Pathologies of EI are visualized and analyzed, and the approximation error between qEI and qLogEI is theoretically bounded. Empirically, LogEI clearly outperforms EI on most tasks, suggesting that it can act as a drop-in replacement for EI.

**Strengths:**

__Good motivation__: Acquisition function optimization is an often-overlooked aspect of Bayesian optimization, and the paper does a good job of displaying the difficulties of acquisition function optimization (Fig. 1) and how the proposed approach remedies the issue.
__Simple, effective and extensible solution:__ Simple solutions that work are great, and LogEI (and its extensions) is good example.
__Very good empirical performance:__ The improvement over EI in the results is striking on most tasks, suggesting it is simply a superior acquisition function to the default.

**Weaknesses:**

__Anecdotal evidence for similarity with EI:__ Intuitively, It is sensible that LogEI is similar to EI. However, there is little evidence (Fig. 1) and theory (Lemma 2) to support this.  I would greatly appreciate a similar Lemma for the analytic variant, and examples of when the two may not be identical. The performance gain of LogEI compared to EI is rather substantial on Ackley (~4x on Ackley-16!) and Mich, which suggests that the two may in fact not be very similar (but that LogEI may in fact simply be superior). Specifically, I don't believe the statement in Row 9, "LogEI, whose members either have _identical or approximately equal optima_ as their canonical counterparts" is well supported.

__Existing LogEI and Lacking references to Related work:__ The idea of a LogEI is not novel (LogEI was an acquisition function in SMAC at one point). Admittedly, that implementation regards log-transformations of the objective [1, 2] and would not help with numerical stability in the same manner. Nevertheless, I think these warrant citation and  _comparison in the experiments_, given the similarities. Moreover, it limits the novelty of the approach.

__Relevance in high dimensions:__ Currently, I am not convinced by the justification for LogEI in high dimensions. To me, LogEI aims to address the pathology where the acquisition function is zero, but I don't see that happening in high-dimensional problems due to the exuberance of high-uncertainty regions (which would make the acquisition function _non-zero, but constant?_). So, why is the proposed approach particularily important in high dimensions - i.e., why does LogEI help when the acquisition function is constant as opposed to (almost) zero? This would, in my opinion, require a separate motivation than Fig. 1, empirical results aside. Moreover, I think the _zero-value_ versus _constant-value_ distinction is very important, and should be emphasized more.

  With this in mind, I find Fig. 2 striking and odd. 60 data points (which is when almost all points have a zero-valued gradient) on an 8-dimensional problem is not a large amount of data (not even a 2x2x...x2 grid), yet the uncertainty is small to the point of "EI and its gradients become numerically zero across most of the domain"? With all due respect, are the authors sure that this is not _just_ the gradient (and not the function value), or that the model has oddly long lengthscales?

__Minor:__
  - __Noisy tasks:__ Adaptation of LogEI to noisy problem settings are missing
  - __Lack of conventional benchmarks__: As a potential drop-in replacement for EI, seeing its performance of the method on the most conventional low-dimensional tasks (Branin, Hartmanns) would be informative. Moreover, it would be helpful for future benchmarking.

For all of these bullets, I believe that evidence to clarify (and not necessarily disprove) the remarks would substantially strengthen the paper.

[1]. An experimental investigation of model-based parameter optimisation: SPO and beyond. F Hutter, HH Hoos, K Leyton-Brown, Kevin P. Murphy. _GECCO '09: Proceedings of the 11th Annual conference on Genetic and evolutionary computation_. 2009.

[2] Sequential model-based optimization for general algorithm configuration. F Hutter, HH Hoos, K Leyton-Brown. _Learning and Intelligent Optimization: 5th International Conference_. 2011.

LogEI in SMAC: https://github.com/automl/SMAC3/blob/29355618b35dcf4b3ce3e773d633109f036dba17/smac/optimizer/acquisition.py#L503

**Questions:**

- Is there any setting where LogEI can _not_ act as a plug-in replacement for EI, or where performance would be expected to be worse?
- Have the authors experimented with LogEI non-continous search spaces, and if so, what are the findings?

**Limitations:**

Some suggestions for addressing limitations have been stated in the Weaknesses section, but are otherwise adequately addressed.

---

> ### Author Rebuttal · Authors · 2023-08-10
>
> Thank you for your detailed feedback about areas that deserve additional discussion.  We seek to clarify the points raised in the following.
>
> __Equivalence of optima of analytical LogEI and EI__
>
> In the CR, we will clarify this statement through a brief Lemma. If the maximum of EI is greater than 0, LogEI and EI have the same set of maximizers. Furthermore, if $\max_{x \in \mathbb X} EI(x) = 0$, then $\mathbb X = \arg \max_{x \in \mathbb X} EI(x)$. In this case, LogEI is undefined everywhere, so it has no maximizers, which we note would yield the same BO policy as EI (where every point is a maximizer). The Lemma is as follows:
>
> _Lemma_: If $\max_{x \in \mathbb X} EI(x) > 0$, then $\arg \max_{x \in \mathbb X} EI(x) = \arg \max_{x \in \mathbb X, EI(x) > 0} LogEI(x)$.
>
> _Proof_:
> Suppose $\max_{x \in \mathbb X} EI(x) > 0$. Then $\arg \max_{x \in \mathbb X} EI(x) = \arg \max_{x \in \mathbb X, EI(x) > 0} EI(x)$.
> For all $x \in \mathbb X$ such that $EI(x) > 0$,  $LogEI(x)  = \log(EI(x))$. Since $\log$ is monontonic, we have that $\arg \max_{z \in \mathbb R_{>0}} z  = \arg \max_{z \in \mathbb R_{>0}} \log(z)$. Hence, $\arg \max_{x \in \mathbb X, EI(x) > 0} EI(x) = \arg \max_{x \in \mathbb X, EI(x) > 0} LogEI(x)$.
>
> Figure 1 shows that LogEI holds approximately the same value as the logarithm of EI, except in regions where the log of EI is numerically zero.
>
>
>
> __Figure 2, Model quality, and constant vs. zero values__
>
> First, we’d like to clarify the data generating process (DGP) for the training data: Training points are not chosen uniformly at random, but rather, 80% are sampled uniformly at random from the domain, and 20% are sampled according to a MVN centered at the function maximum with a standard deviation of 25% of the length of the domain. The idea behind this DGP is to mimic the kind of data one would see during a BO loop (for illustration purposes, without having to run thousands of BO loops to generate the figure). We will clarify this in the CR.
>
> Figure 2 in the MT reflects the fact that under this data generating process with the chosen test problem, the incumbent (best observed point) is much better than the values at the random test locations, and this becomes increasingly the case as the dimensionality of the problem increases and the number of training points grows. For a particular replicate, Figure 1 in the attached PDF shows the model fits in-sample (black), out-of-sample (blue), and the best point identified so far (red) for our DGP with 60 training points and a random subset of 50 (out of 2000) test points. One can see that the model produces decent mean predictions for out-of-sample data, and that the uncertainty estimates appear reasonably well-calibrated (e.g., the credible intervals typically cover the true value). Because of this, we do not see any clear evidence that there is something odd about the model and its length scales.
>
> What Figure 1 in the attached PDF does show is that while there is ample uncertainty in the predictions of the model away from the training points, for the vast majority of points, the mean prediction is many standard deviations away from the incumbent value (the error bars are +/- 2 standard deviations). This is the key reason for EI taking on zero (or vanishingly small) values and having vanishing gradients.
>
> To illustrate this, Figure 2 in the attached PDF shows the histogram of `z(x)` values, the argument to the function `h` in eq (2). It also contains the thresholds corresponding to the values `z` below which `h(z)` is less than the respective threshold. Since `sigma(x)` is close to 1 for most test points (mean: 0.87, std: 0.07), this is more or less the same as saying that `EI(z(x))` is less than the threshold. It is evident from the histogram that the majority of the test points fall below these threshold values (especially for larger thresholds), showing that the associated acquisition function values (and similarly the gradients) are numerically almost zero and causing issues during acquisition function optimization.
>
> __Relevance in high dimensions__
>
> LogEI tends to be particularly effective in high dimensions and we can see how the discussion of Lemma 1 did not adequately address this.
>
> It is important to note that dimensionality alone is not the key factor at play, but the measure (“volume”) of points that attain significantly sub-optimal objective values in the search space (the right-hand side of Lemma 1).
>
> While the average posterior uncertainties tend to be larger in a larger proportion of the search space as  the dimensionality increases, they are usually much smaller than the empirical standard deviation of the objective values, and this means that the predictive mean can still be many predictive standard deviations away from the incumbent value for large swaths of the search space.
>
> Fig. 1 and 2 in the attached pdf are an empirical validation of this intuition: while the posterior uncertainties are large and cover the errors, the corresponding EI values are vanishingly small.
>
> __Existing work__
>
> As you already noted in your review, SMAC’s “LogEI” acquisition function tackles a fundamentally different problem than our LogEI. In the former, the surrogate model is fit to log-transformed outcomes in the hope that a log-transformation improves the model fit and optimization performance. Despite the nominal similarity, transformations of the outcomes before fitting the surrogate model are entirely orthogonal to this work, as they do not seek to solve the fundamental problem of the acquisition function itself being hard to optimize due to vanishing gradients. In fact, the techniques underlying our LogEI acquisition function are complementary to the outcome transformation (e.g. a log transform in SMAC’s LogEI) and would help resolve numerical issues and vanishing gradients. We will clarify these differences in the section on related work. See Fig. 4 in the PDF for a comparison of LogEI with SMAC's LogEI.

---

> > ### Comment · Reviewer_u8yk · 2023-08-10
> > **Initial response**
> >
> > Thanks for the additional plots. These really strengthen the motivation for the paper, and (in my opinion) highlight the vanishing gradient even better.
> >
> > An aside:
> > _I found myself checking the exact value at which torch.float64 (and torch.float32) rounds to zero, and thereafter, exactly how many standard deviations of tolerance EI supports. Such reference values (in both Rebutal Fig. 1 and Fig. 2) are, in my view, even more informative than the provided thresholds._
> >
> > __Existing work__
> >
> > I agree with the authors on this point. The existence of SMAC LogEI does not limit the novelty. Thanks for including it in the rebuttal plots.
> >
> >
> >
> > __Equivalence of optima of analytical LogEI and EI:__
> >
> > The extraordinary difference between LogEI and EI is unsettling to me, especially given the Lemma for the CR. This, to me, should mean one or two (or both) things:
> > 1. This method is very, _very_ necessary since EI truly is zero just about everywhere
> > 2. Not enough budget is allocated towards optimizing EI
> >
> > While I am _decidedly not_ advocating for simply increasing the budget of EI, it seems like the gap is simply too large at the moment. Since the lemma suggests that an increased budget is guaranteed to solve the problem, are the authors able to provide an ablation as to when this happens (or at least, provide a sense of the rate)?
> >
> > For now, I have increased my score to a 6. To further increase it (which I am willing to do) I would appreciate the aforementioned ablation to assess the practical impact on the budget allocated to BO acquisition function maximization.

---

> > > ### Author Response · Authors · 2023-08-12
> > >
> > > Thank you for your thoughtful questions and interest.
> > >
> > > __Ablations on initialization heuristics__
> > > > I would appreciate the aforementioned ablation to assess the practical impact on the budget allocated to BO acquisition function maximization.
> > >
> > > We believe the ablations in the last two figures (Fig. 17 and Fig. 18) of the Appendix of our submission can clarify this.
> > > - Fig. 18 shows the regret of q(Log)EI on the 16-dimensional Ackley and Levy test problems using 1, 4, and 16 random restarts, and for q = 1, 4, and 16. For the Ackley q = 1 case, we see that increasing the number of restarts does improve the performance of qEI. However, extrapolating from the small increase in performance from 4 to 16 restarts, it seems exceedingly unlikely one could match the performance of qLogEI within a practical compute budget by scaling up the number of restarts. On Levy q = 1 on the other hand, the performance of qEI and qLogEI is similar. This is because the distribution of near-optimal values (relevant for the right-hand side of Lemma 1) is much less peaked around the optimal input for Levy, than it is for Ackley (see plots of Levy and Ackley for a visual illustration).
> > >
> > > - Fig. 17 displays an ablation on the impact of initialization heuristics, comparing random restarts with BoTorch’s default Boltzmann-sampling-based approach. One can see that BoTorch’s initialization heuristic helps ameliorate the performance of canonical EI somewhat, but in no way closes the performance gap to LogEI. While prior research has produced many more initialization heuristics, as we discuss in Appendix B.1, they do not resolve the fundamental issues in computing EI that are addressed here.
> > >
> > > __Numerical thresholds__
> > > > An aside: I found myself checking the exact value at which torch.float64 (and torch.float32) rounds to zero, and thereafter, exactly how many standard deviations of tolerance EI supports. Such reference values (in both Rebutal Fig. 1 and Fig. 2) are, in my view, even more informative than the provided thresholds.
> > >
> > > That's a good point. It is important to clarify that there are at least two numerical threshold that lead to pathologies:
> > > 1) Numerical underflow: $x = 0$ numerically, but $x \neq 0$ mathematically.
> > > 2) Numerical precision: $1+x = 1$ numerically, but $x \neq 0$ numerically.
> > >
> > > Underflow (1) implies that both the value and gradient of standard EI is numerically exactly zero.
> > >
> > > Numerical precision (2) plays a key role in gradient-based optimization, where the parameters are incremented by a scaled gradient $x_{n+1} = x_n - \alpha \nabla f(x_n)$ and $\alpha$ is the step size. If $\alpha \nabla f(x_n)$ in this expression becomes smaller than the numerical precision ($\approx$ 1e-8 for single and $\approx$ 1e-16 for double precision floating point numbers), the gradient increment is likely to be a no-op, i.e. $x_{n+1} = x_n$ numerically, even if $\alpha \nabla f(x_n) \neq 0$ *numerically*.
> > >
> > > For quasi-second order methods like L-BFGS, which we used for the experimental results, the gradient is further scaled by an approximation to the inverse Hessian, which makes reasoning about the precise thresholds more involved, but the fundamental issue remains the same. The optimization step becomes a no-op if the *(step size + inverse Hessian)-scaled* gradient value is below numerical precision, which is increasingly likely to happen for the thresholds we indicated.
> > >
> > > Most practical implementations further use non-zero convergence tolerance parameters that trigger the optimizer to terminate when the gradient magnitude is sufficiently small. In order not to conflate these effects, we did not use any (non-zero) convergence tolerance parameters for the experiments of this paper.
> > >
> > > We will add this elaboration.

---

> > > > ### Comment · Reviewer_u8yk · 2023-08-14
> > > > **Thank you!**
> > > >
> > > > Thanks to the authors for directing me towards Fig. 17 and 18 - these figures were indeed what I was looking for. Moreover, I appreciate the additional explanation.
> > > >
> > > > I am very happy for the authors' additional clarifications, and have thus increased my score to a 7.

---

> > > > > ### Author Response · Authors · 2023-08-14
> > > > >
> > > > > We are glad to hear that!

---

### Official Review · Reviewer_o9Cr · 2023-07-04

**Soundness:** 4 excellent
**Presentation:** 4 excellent
**Contribution:** 4 excellent
**Rating:** 7
**Confidence:** 4

**Summary:**

This paper identifies a numerical pathology with the expected improvement (EI) family of acquisition functions: the vanishing gradients of the acquisition function leads to failure in acquisition function optimization. A set of modified EI acquisition functions that fix the numerical pathologies have been proposed. In experiments, the proposed EI acquisition functions are shown to outperform the canonical EI acquisition functions and to perform on par with other state-of-the-art acquisition functions on Bayesian optimization benchmarks.


**Strengths:**

- This paper focuses on a previous neglected aspect of Bayesian optimization, acquisition function optimization, and identifies the numerical pathologies associated with EI acquisition functions. This paper provides a theoretical analysis of the vanishing gradient issue.
- Improvements have been proposed to EI, Monto Carlo Parallel EI, Constrained EI and EHVI.
- Experiments clearly show the numerical pathology of EI optimization and the superior performance of the improved EI version.
- The proposed improved EI acquisition functions perform on par with state-of-the-art acquisition functions on high dimensional synthetic functions in both sequential and batch settings.


**Weaknesses:**

- The proposed treatment only works for the acquisition functions with vanishing gradient issues.


**Questions:**

- It is great to have an error bound of the qLogEI. However, for acquisition functions, preserving the relative order of values is more important than absolute difference. I wonder to what extent qLogEI preserves the relative order of values compared to qEI.



**Limitations:**

The limitation of the proposed method has been discussed in the last section.

---

> ### Author Rebuttal · Authors · 2023-08-10
>
> Thank you for your encouraging your review.
>
> > It is great to have an error bound of the qLogEI. However, for acquisition functions, preserving the relative order of values is more important than absolute difference. I wonder to what extent qLogEI preserves the relative order of values compared to qEI.
>
> While the smooth approximation to the ReLU in the integrand of qEI is monotonically increasing and upper-bounds the ReLU, the smooth approximation to the max operator can change the relative ordering. Therefore, while the relative ordering is fully maintained for the analytical LogEI version, this cannot generally be guaranteed for the Monte-Carlo (batch) version.
>
> Nevertheless, the absolute error bound guarantees that the maximizer of qLogEI attains a similar acquisition value to the maximizer of qEI. Whether or not that is a similar point in the input space depends on the surrogate model during the particular iteration. If the EI acquisition value is a good indicator of optimization performance, such a change in ordering would not lead to a significant decrease in optimization performance, and if EI is not a good indicator of optimization performance on a particular problem, it would not be recommended to use EI in any case.
>
> Notably, our empirical experiments demonstrate that the benefits of a smooth acquisition optimization landscape with strong gradients far outweigh the potential for a small change in relative ordering of points with promising acquisition values.

---

> > ### Comment · Reviewer_o9Cr · 2023-08-16
> >
> > Thanks for the reply. I understand a guarantee for relative ordering is hard, especially with Monte Carlo approximation. The response makes sense. It address my concern. Overall, the paper presents a nice tricky that addresses a real world challenge of qEI.

---

### Official Review · Reviewer_QK5L · 2023-07-06

**Soundness:** 3 good
**Presentation:** 3 good
**Contribution:** 3 good
**Rating:** 7
**Confidence:** 3

**Summary:**

In this paper, the authors identify both through examples and theoretical analysis several numerical pathologies inherent to the computation and optimization of the Expected Improvement (EI), a popular acquisition function at the heart of Bayesian Optimization (BO) algorithms.
They subsequently propose a numerical reformulation, LogEI, which achieves substantially better performances on a quite extensive range of benchmarks. This reformulation applies to all the member of the EI family: constrained EI for constrained BO, parallel EI for batch BO, and expected hypervolume improvement for multi-objective BO.

**Strengths:**

- The paper is well-written and well-organised.
- The proposed numerical fix for EI is likely to have a great impact as it may benefit to all public implementations of EI. Furthermore, it does not incur excessively longer computation times except perhaps for multi-objective BO (roughly one order of magnitude larger).
- LogEI seems to produce more consistent results with respect to the initial optimization starting point compared to its canonical counterpart EI, thus reducing the need for heuristics for that matter.
- All claims are backed up by an impressive amount of numerical experiments and ablation studies in each setting (vanilla BO/ constrained BO/ batch BO/multi-objective BO).

**Weaknesses:**

I did not spot any weakness.

**Questions:**

I do not have any questions.

Edit: I have read the rebuttal and the discussions between the authors and other reviewers, my score remains unchanged.

**Limitations:**

None.

---

> ### Author Rebuttal · Authors · 2023-08-10
>
> We thank you for your positive review.

---

### Author Rebuttal · Authors · 2023-08-10

We thank the reviewers for their detailed and predominantly positive reviews. We are attaching a one-page pdf with additional figures to help answer questions that arose during the review process, and are responding to each reviewer's questions in detail below and in the comments.

__Generality of the Methodology__

> The proposed treatment only works for the acquisition functions with vanishing gradient issues.

Our work brings to light the fundamental challenges of optimizing the popular EI family of acquisition functions (AFs), and proposes remediations tailored to the analytic, batch, constrained, and multi-objective setting, which cover extensive application domains. Through the lens of this popular family of AFs, we investigate how the formulation of AFs contribute to one’s ability to optimize them, and in general how experimental results may be subject to the quality of the AF optimization procedure. While we consider a broad class of EI-based acquisition functions in this work, we hope that it can inspire further investigation, thought, and potential remediation in the development of other types of acquisition functions, promising examples including entropy-based acquisition functions.

__Extension to Noisy Observations__

We point out that LogEI extends naturally to the Noisy Expected Improvement (NEI) acquisition function. We did not include NEI in our original manuscript since the construction is a trivial extension of qEI. The only difference between qEI and qNEI is that the computation of "best_f" values are based on samples of the GP at previously observed points, rather than taking the empirically observed objective value (cf. Balandat et al. NeurIPS 2020, S5.2). We then directly use the forward pass of qLogEI to compute qLogNEI.

In the attached PDF (Fig. 3), we show empirical optimization performance of qLogNEI compared to qLogEI, EI, NEI, and Gibbon on Hartmann 6D, Ackley 8D, and Ackley 16D for varying noise levels. We set the noise level as a proportion of the total range of the respective function, which is ~3.2 for Hartmann and ~20 for Ackley. Thus, a noise level of 1% * Range(f) is equivalent to Gaussian noise with a std of 0.2 for Ackley.

On Hartmann 6D, qNEI, qLogNEI, and Gibbon consistently find the optimum in the allocated number of iterations. qEI and qLogEI exhibit higher variance in their optimization traces especially at the highest noise level, which is expected since the `best_f` value they rely on becomes highly stochastic and might be far from the true (noiseless) best objective value corresponding to the queried inputs.

Notably, qLogNEI outperforms both canonical EI counterparts and leads to significantly improved optimization on Ackley. qLogNEI also leads to notable improvements over Gibbon for the higher-dimensional functions.

We will include a description of qLogNEI, include its implementation in the code release, and add a treatment of these results in the SM.

__Non-continuous search spaces__

Our work largely focuses on the problem of vanishing gradients, which is most pronounced for problems involving continuous or mixed spaces.  We expect the proposed methods to have less impact in fully discrete or mixed spaces, especially when some combinations of input parameters can be exhausted entirely. Nevertheless, since our work ensures that the acquisition function values (not just the gradients) do not numerically become zero, it will result in better optimization performance in settings where the feasible choices are all “far” from the incumbent. Promising approaches to gradient-based optimization of fully-discrete or mixed spaces with difficult-to-enumerate combinations such as Probabilistic Reparameterization or straight-through estimators (Daulton et al. NeurIPS 2022) may particularly benefit from LogEI.  We will add a discussion of this to the CR.

__Conventional Benchmarks__

As a potential drop-in replacement for EI, seeing its performance of the method on the most conventional low-dimensional tasks (Branin, Hartmanns) would be informative. Moreover, it would be helpful for future benchmarking.

We included benchmarks that had flexible numbers of input dimensions, many constraints, or many outcomes to highlight how these factors impact the vanishing gradient problem and benefit from LogEI.  We agree that including such canonical test problems are worth documenting and will include them in the CR. In the interim, please see the attached PDF for the results on Hartmann and Branin (Fig. 4). LogEI outperforms alternatives on Hartmann (6 dimensional) and matches the performance of EI on Branin (2 dimensional).

__LogEI as Replacement of EI__

LogEI should work for any case where EI works, as the two acquisition functions have the same optima (in the analytic case) — LogEI just makes optimizing for those optima much easier. One could imagine cases where the model is incorrect or so degenerate that other model-free policies such as random search or evolutionary strategies work better.  To our knowledge, such concerns are true of most model-based acquisition functions studied in the literature. Otherwise, we confidently recommend LogEI as a drop-in replacement of EI.
On a practical note, our implementations indeed already share the same interface as BoTorch’s EI variants, making such a replacement particularly convenient in one popular BO framework.

---

### Decision · Program_Chairs · 2023-09-21

**Decision:**

Accept (spotlight)

**Comment:**

The reviewers and meta reviewer all appreciated the quality of the work with a clear, well-written manuscript. The paper focuses on a simple trick (a drop-in replacement) to improve the widely-used expected-improvement acquisition function. The reviewers and meta reviewer are convinced that this paper will inform future research and the work of practitioners.

They thank the authors for their response and their efforts during the rebuttal phase, which was helpful to improve the submission and clarify concerns (e.g., relation to previous work). The reviewers and meta reviewer unanimously recommend the paper for acceptance.